# ADVERSARIAL DETECTOR FOR DECISION TREES ENSEMBLES USING REPRESENTATION LEARNING

## ABSTRACT

Research on adversarial evasion attacks focuses mainly on neural network models. Among other reasons, this is because of their popularity in certain fields (e.g., computer vision and NLP) and the models' properties, making it easier to search for adversarial examples with minimal input change. Decision trees and tree ensembles are still very popular due to their high performance in fields dominated by tabular data and their explainability. In recent years, several works have defined new adversarial attacks targeting decision trees and tree ensembles. As a result, several papers were published focusing on robust versions of tree ensembles. This research aims to create an adversarial detector for attacks on an ensemble of decision trees. While several previous works have demonstrated the generation of more robust tree ensembles, the process of considering evasion attacks during ensemble generation can affect model performance. We demonstrate a method to detect adversarial samples without affecting either the target model structure or its original performance. We showed that by using representation learning based on the structure of the trees, we achieved better detection rates than the state-of-the-art technique and better than using the original representation of the dataset to train an adversarial detector.

## 1 INTRODUCTION

In recent decades we have seen the introduction of machine learning algorithms in production environments into various fields such as medical imaging (Zhou et al., 2021), autonomous driving (Huang & Chen, 2020) and law enforcement (Vestby & Vestby, 2019). With the leap in performance of those models and their integration into real-life systems, people began to investigate how to bypass classifiers and defend against those malicious attempts (Dalvi et al., 2004; Lowd & Meek, 2005).

Many papers have addressed examples of adversarial attacks that make small changes that are hard for a human to notice in the inputs of a machine learning model, usually a neural network, so their predictions are wrong. These can be exploited by a malicious actor and used to bypass a model that might, for example, be responsible for a critical classification task affecting people's lives. As a result, various researchers published techniques to detect and defend against adversarial attempts. Most of the research is focused on adversarial attacks targeting neural network models, among other things, because of the nature of their continuous learning space, which allows a gradient ascent process to maximize the model's loss function given a specific input. Thus defenses and detectors mainly target neural network models as well.

Tree-based models continue to be very popular, especially for tabular data tasks (Nielsen, 2016; Shwartz-Ziv & Armon, 2022; Grinsztajn et al., 2022), because they usually demand less data and are more interpretable. There are fewer studies on adversarial attacks and defenses affecting decision tree models. Gradient-descent-based methods commonly used in earlier attack models cannot be applied directly to evade decision trees due to the discrete nature of their non-differentiable decision-making paths and tree-splitting rules. Unfortunately, this does not mean that decision trees are unaffected by evasion attacks.

In this work, we present a detection technique for adversarial evasion attacks against tree-based classifiers, focusing on boosting ensembles. Our main contributions are: (*i*) We defined a task that allows us to generate sample representations that rely on the distribution of the dataset in the

different routes of a tree ensemble. *(ii)* We designed a pipeline to train and evaluate adversarial detection with reduced possibilities of overfitting or bias.

## 2 MOTIVATION

Chen et al. (2019) proposed a robust decision trees technique against adversarial evasion attacks. The model training algorithm was changed, as a result of which model itself was changed. As part of the experiment, the new model's accuracy was checked and compared to the non-robust model. Of the eleven datasets tested, seven showed a decrease in accuracy.

Our primary motivation for this work is to create a defense layer for a decision tree ensemble against adversarial attacks. Our defense layer does not affect the model itself, allowing the model owner to decide if they want defense applied to their existing system.

Secondly, production tree-based models use well-known open-source libraries such as XGBoost (Chen & Guestrin, 2016b), CatBoost (Dorogush et al., 2018), and LightGBM (Ke et al., 2017). These libraries are heavily used, tested, and improved, which is partly why they were chosen in the first place. Currently, as of writing this paper, the above libraries do not contain an official version that is robust against adversarial attacks. Therefore, to add adversarial robustness to a model, it is necessary to use a different third-party version of the model or develop a new one.

## 3 BACKGROUND

### 3.1 RELATED WORK

We can split the field of adversarial learning into three primary sectors: attacking methods, defending methods, and detectors which aim to detect whether or not a sample is adversarial without changing the model itself.

**Generating Adversarial Samples.** Early work around generating adversarial samples (Goodfellow et al., 2014; Kurakin et al., 2016) used backpropagation to try and discover which input features we should change to maximize the loss function of a model. In the face of more recent attacks, a different loss functions was suggested to find an adversarial sample (Papernot et al., 2016b; Carlini & Wagner, 2017; Cheng et al., 2018). Other works used concepts from geometry and the location of the boundaries between decision spaces to search for a minimal perturbation for creating an adversarial sample (Moosavi-Dezfooli et al., 2016; Yang et al., 2020).

Because decision tree classifiers are not a continuous space model, earlier backpropagation methods will not work in these cases. Black-box methods, which ignore the internal inner structure of the model, try to approximate the gradients and can generate attacks for decision-trees-based classifiers using multiple queries to find the boundary between different classes in the unknown decision space (Cheng et al., 2018; Chen et al., 2020).

Some relevant white-box techniques focus specifically on the nature of decision trees (Papernot et al., 2016a; Kantchelian et al., 2016; Zhang et al., 2020). Papernot et al. (2016a) defined an algorithm to search for a given sample, the closest leaf in the neighborhood of the original leaf, and perturb the features between them. Kantchelian et al. (2016) formulated a set of equality and inequality constraints based on the tree structure to generate an optimal adversarial sample for tree ensembles using a mixed-integer linear program.

**Model Defenses Against Adversarial Attacks.** Common approach for protecting models is to train a robust model for evasion attacks. Adversarial training (Goodfellow et al., 2014) is one of these methods, with which one can generate adversarial samples and add them to the training data. Other suggested solutions use known techniques with other purposes, such as knowledge distillation, as shown in Papernot et al. (2016c), which used its traits to create a newer model version with smaller gradients to make it more difficult to generate adversarial samples. Another work is Wang et al. (2018), which used dropout in prediction time to reduce the dependency on specific neurons in a neural network.

Defending decision tree classifiers combine the structure of the trees with the methods mentioned above. Adversarial boosting was suggested in Kurakin et al. (2016) with an idea similar to adversarial training, and in each boosting round, adversarial samples are created and added to the next round's training data. Other works show how to generate more robust tree models with new optimization formulations while considering evasion attacks and perturbations (Chen et al., 2019; Andriushchenko & Hein, 2019; Calzavara et al., 2020; Vos & Verwer, 2021; 2022).

**Adversarial Detectors.** There are two main approaches when building an adversarial attack detector: using statistics and hypothesis testing to investigate if there is any difference between the regular samples distribution and the adversarial sample's distribution (Feinman et al., 2017; Grosse et al., 2017; Katzir & Elovici, 2019), and training a machine learning classifier to act as a detector of adversarial attempts (Metzen et al., 2017; Fidel et al., 2020). Recent work used the combinations of leaves in tree ensemble predictions, called output configurations, to detect abnormal leaves configuration of a sample compared to a reference dataset to detect an adversarial sample using a defined metric called OC-score (Devos et al., 2022). We will compare our work to Devos et al. (2022), which is considered the state-of-the-art in detecting adversarial samples on decision trees at the time of writing this paper.

## 3.2 PROBLEM FORMULATION

For a given classifier $C$, a given sample $x$, with set of features $F$ and a label $y$ in which $C(x) = y$, an adversarial sample $x'$ is defined using an adversarial perturbation $\delta : x' = x + \delta$. An adversarial sample can be generated by a targeted or untargeted attack. For an untargeted attack we want to find $\delta$ that meet with the condition:

$$C(x') \neq y \quad \text{s.t.} \quad ||\delta||_p < \epsilon \tag{1}$$

Which means that $C(x) \neq C(x')$ where the $p$-norm of $\delta$ will be limited by a value $\epsilon$. For a targeted attack we need to define a target class $t$ where $t \neq y$ and:

$$C(x') = t \quad \text{s.t.} \quad ||\delta||_p < \epsilon \tag{2}$$

For $p \in \mathbb{N}_1$ (All natural numbers without zero) the $p$-norm is defined: $||\delta||_p := (\Sigma_{i=1}^{F}(\delta_i)^p)^{\frac{1}{p}})$

For $p = \infty$ - measures the largest absolute difference between two features and is defined: $||\delta||_\infty = \max_{x_i} |x_i - x'_i|$

Given a decision tree model $\mathcal{T}$ and a sample $x$, our classification task is to detect whether the sample is normal or is an attempted adversarial attack.

## 4 METHOD

### 4.1 METHOD GENERAL FLOW

Our method consists of 11 main steps:

1. Split the dataset into four different parts for different purposes: $\mathcal{S}_\mathcal{T}$ to train the tree model, $\mathcal{S}_\mathcal{E}$ to train our basic representation model, $\mathcal{S}_{\mathcal{D}-train}$ to train our adversarial detector and $\mathcal{S}_{\mathcal{D}-test}$ to evaluate our adversarial detector.

2. Train a tree model.

3. Generate a triplets dataset that will be used to initialize the new representations. This is explained more fully in Subsection 4.2.

4. Train our basic embedding model $\mathcal{E}$. This is explained more fully in Subsection 4.3.

5. Split $\mathcal{S}_{\mathcal{D}-train}$ and $\mathcal{S}_{\mathcal{D}-test}$ into two parts.

6. Generate adversarial samples using an attack method $\mathcal{A}$.

7. Generate a new triplet dataset for each of the new sub-datasets. This is explained more fully in Subsection 4.4.

8. Optimize the representations of the new sub-datasets to our new embeddings using $\mathcal{E}$, more details in Subsection 4.4.

9. Concatenate the new representation of every set to the original ones.

10. Train our adversarial detector. This is explained more fully in Subsection 4.5.

11. Evaluate our adversarial detector. This is explained more fully in Section 5.

A detailed visualization sketch shown in Figure 28 in Appendix H, together with a further explanation about the dataset splitting.

## 4.2 Dataset Representation

At the heart of our method is the idea that we want to extract a new representation of a dataset based on the structure of a target tree ensemble model to understand the behavior of normal samples and detect adversarial samples. To extract a meaningful representation, we took inspiration from an embedding process. For each sample, we assign a vector of random numbers with $d$ dimensions, called latent features, that will be optimized using a gradient descent process using a simple feed-forward neural network based on the structure and traits of the tree.

For a given dataset $\mathcal{S}$ and a trees model $\mathcal{T}$ we define a new dataset:

$$\mathcal{R}_{\mathcal{S}}^{\mathcal{T}} = \{(s_i, s_j, n_k) | s_i, s_j \in \mathcal{S}, n_k \in \mathcal{N}_{\mathcal{S}}^{\mathcal{T}}, s_i \in n_k \wedge s_j \in n_k\} \tag{3}$$

Where $\mathcal{N}_{\mathcal{S}}^{\mathcal{T}}$ is the set of all internal nodes of $\mathcal{T}$ reached by at least one sample from $\mathcal{S}$, $n$ is a single node, and $s$ is a single sample. Each triplet in $\mathcal{R}_{\mathcal{S}}^{\mathcal{T}}$ contains two samples and a node. Both samples reach the internal node $n_k$ in $\mathcal{T}$. We define the supervised task below:

$$f(s_i, s_j, n_k) = \begin{cases} 1 & \text{if } s_i \text{ and } s_j \text{ pass to the same child of } n_k \\ 0 & \text{otherwise} \end{cases} \tag{4}$$

To collect this new dataset, we take our original dataset and traverse with each of the samples through the different routes in the trees in the ensemble and save the routes aside. During the process, we document which sample passes in which node and create the mapping:

$$\mathcal{M}_{\mathcal{S}}^{\mathcal{T}}(n_k) = \mathcal{S}_{n_k} \tag{5}$$

Which returns for a given node all the samples which reached it during the above traverse. Then to generate $\mathcal{R}_{\mathcal{S}}^{\mathcal{T}}$ we can use Algorithm 1.

---

**Algorithm 1** Generating triplets dataset to train the basic embedding model to fine-tune future representations

**input** : set of nodes $\mathcal{N}_{\mathcal{S}}^{\mathcal{T}}$, model $\mathcal{T}$, nodes to samples mapping $\mathcal{M}_{\mathcal{S}}^{\mathcal{T}}$, size of final dataset N
**output:** $\mathcal{R}_{\mathcal{S}}^{\mathcal{T}}$, labels

1: **for** $i = 1$ to N **do**
2:     Sample a random node $n_k$ from $\mathcal{N}_{\mathcal{S}}^{\mathcal{T}}$
3:     Sample 2 random samples $s_i, s_j$ from $\mathcal{M}_{\mathcal{S}}^{\mathcal{T}}(n_k)$
4:     $\text{triplet}_i \leftarrow (s_i, s_j, n_k)$
5:     **if** $s_i$ and $s_j$ agree on the condition in $n_k$ **then**
6:         $\text{label}_i \leftarrow 1$
7:     **else**
8:         $\text{label}_i \leftarrow 0$
9:     **end if**
10: **end for**

---

Algorithm 1 returns a list of triplets constructed from 2 samples and one node and a list of labels based on the above logic. For a chosen N, which is the final size we choose for the dataset $\mathcal{R}_{\mathcal{S}}^{\mathcal{T}}$, we generate the triplets described above. After sampling for a node and relevant samples, in line 5, we check if both samples agree on the condition of the feature threshold in $n_k$, in which case they move to the same child on $n_k$, and the label of the new sample is set to 1. Otherwise, it is set to 0.

### 4.3 EMBEDDING MODEL

To generate our The basic embedding model is trained using $\mathcal{R}^{\mathcal{T}}_{\mathcal{S}_{\mathcal{E}}}$. We can see in Figure 1-(a) a general sketch of the architecture we used. During the embedding model-training phase, we use two embedding matrices, one for the samples and one for the nodes, and we optimize the representation with a gradient descent process based on the task defined earlier.

Each sample vector is initiated with a random vector of dimension $d_s$ and each node with a random vector of dimension $d_n$. The vectors of each sample and the one representing the node are concatenated and passed through a single feed-forward layer with a ReLU activation function and finish in a feed-forward layer with a sigmoid activation function used to evaluate the loss value. The model is trained with a binary cross-entropy loss function and an Adam optimizer. Then, the model weights will be saved to optimize the new sample's representations.

### 4.4 NEW SAMPLES EMBEDDING

When we want to extract the representation of new samples set $\mathcal{S}_{new}$, we first create a new dataset:

$$\mathcal{R}^{\mathcal{T}}_{\mathcal{S}_{\mathcal{E}},\mathcal{S}_{new}} = \{(s_i, s_j, n_k)|s_i \in \mathcal{S}_{\mathcal{E}}, s_j \in \mathcal{S}_{new}, n_k \in \mathcal{N}^{\mathcal{T}}_{\mathcal{S}_{new}}, s_i \in n_k \wedge s_j \in n_k\} \tag{6}$$

This means the triplet in this new dataset is constructed from one sample from the samples used to train the embedding model ($\mathcal{S}_{\mathcal{E}}$) and another sample from the new sample set whose representation we want to optimize. The labels are set in the same manner as we described before, based on the fact that the two samples align with the feature threshold in the relevant node. We use algorithm 2 to construct this dataset.

---

**Algorithm 2** Generating triplets dataset from a new sample set to optimize its new representations

---

**input** : set of nodes $\mathcal{N}^{\mathcal{T}}_{\mathcal{S}_{new}}$, model $\mathcal{T}$, a mapping between nodes to samples which were used to train the embedding model $\mathcal{M}^{\mathcal{T}}_{\mathcal{S}_{\mathcal{E}}}$, mapping between nodes to samples from the new dataset $\mathcal{M}^{\mathcal{T}}_{\mathcal{S}_{new}}$, size of final dataset N

**output:** $\mathcal{R}^{\mathcal{T}}_{\mathcal{S}_{\mathcal{E}},\mathcal{S}_{new}}$, labels

1: **for** $i = 1$ to N **do**
2:    Sample a random node $n_k$ from $\mathcal{N}^{\mathcal{T}}_{\mathcal{S}_{new}}$
3:    Sample a random samples $s_i$ from $\mathcal{M}^{\mathcal{T}}_{\mathcal{S}_{\mathcal{E}}}(n_k)$
4:    Sample a random samples $s_j$ from $\mathcal{M}^{\mathcal{T}}_{\mathcal{S}_{new}}(n_k)$
5:    triplet$_i \leftarrow (s_i, s_j, n_k)$
6:    **if** $s_i$ and $s_j$ agree on the condition in $n_k$ **then**
7:       label$_i \leftarrow 1$
8:    **else**
9:       label$_i \leftarrow 0$
10:   **end if**
11: **end for**

---

Similarly to Algorithm 1, in Algorithm 2 we choose N, which is the size we choose for the dataset $\mathcal{R}^{\mathcal{T}}_{\mathcal{S}_{\mathcal{E}},\mathcal{S}_{new}}$. In line 3, we take random samples from $\mathcal{M}^{\mathcal{T}}_{\mathcal{S}}(n_k)$, a sample that was used to train the basic representations that reached $n_k$. In line 4, we sample random samples from $\mathcal{M}^{\mathcal{T}}_{\mathcal{S}_{new}}(n_k)$ which is a sample from the new dataset whose representation we want to optimize.

Then, in line 6, we check if both samples agree on the condition of the feature threshold in $n_k$, in which case they move to the same child of $n_k$, and the label of the new sample is set to 1. Otherwise, it is set to 0. Algorithm 2 returns a list of triplets constructed from two samples and one node and a list of labels.

Then the weights of the trained embedding model are loaded to the same architecture with all weights frozen (both layers' weights and biases, embedding matrix of the $\mathcal{S}_{\mathcal{E}}$ samples and embedding matrix of the nodes), and a new unfrozen embedding matrix for the new samples is initialized as described in Figure 1-(b).

The representations are then optimized with a gradient descent process while using $\mathcal{R}^{\mathcal{T}}_{\mathcal{S}_{\mathcal{E}}, \mathcal{S}_{new}}$.

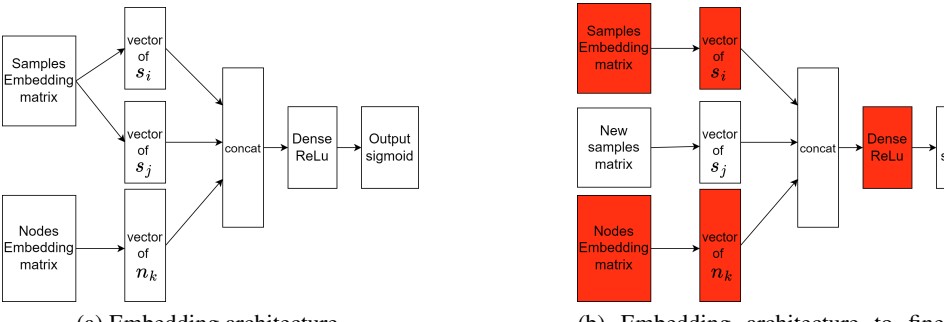

(a) Embedding architecture.

(b) Embedding architecture to fine-tune new samples representations.

Figure 1: (a) - General architecture sketch of embedding model. (b) - changes were done to fine-tune the new samples' representations. The elements in red are frozen weights and biases which remain unchanged during a gradient descent process.

### 4.5 ADVERSARIAL CLASSIFIER

As suggested by Metzen et al. (2017), we trained an adversarial samples detector based on a classifier to try and classify whether a specific sample is a normal sample that came from the original distribution of the input samples of the original dataset or an adversarial sample. We extract four new sub-datasets from the datasets used for training and evaluating the detector. We extract two from each one, a set that stayed as the normal samples, and using the other set, we generate adversarial samples and then throw the original samples away. Afterward, we extracted each dataset's new representations using the process described in Subsection 4.4. We used an XGBoost as the classifier of the detector. As an input to the classifier, we concatenated our new extracted representations to the original features and used them together as the final features set.

## 5 EVALUATION

### 5.1 EXPERIMENTAL SETUP

In our experiments, we tested the performance of our method of creating new representations by training an adversarial-evasion-attack detector on 18 datasets, described in Subsection 5.2, with five different attacks, described in Subsection 5.3. We compared our method to OC-score (Devos et al., 2022), considered the state-of-the-art for that task, and to detector classifiers that we trained on the original representations of the datasets. We tested our method against two tree-based ensembles: XGBoost and RandomForest - both implemented by Chen & Guestrin (2016a). The full results tables for XGBoost are given in Appendix B and for RandomForest in Appendix C. To train our adversarial classifier, we used a second XGBoost model; our positive samples are the adversarial samples, and the negative samples are the normal ones. To train and test our detector, we used no more than 100 samples for black-box attacks and 1000 for the white-box attack from $\mathcal{S}_{\mathcal{D}-train}$ and $\mathcal{S}_{\mathcal{D}-test}$. While generating adversarial samples, we only attacked using samples originally classified correctly by the target model. For the embedding dimensions we chose $d_s = 250$ and $d_n = 100$. We used ROC-AUC and PR-AUC as our metrics. For multi-class datasets, each metric is calculated for each of the different labels in a 1-vs-all manner, and eventually, an average is calculated. Our code is online in github[1].

### 5.2 EVALUATED DATASETS

In our experiments, we tested with 18 classification datasets, some of them binary and others multi-class. The datasets are described in Table 1. The top section of Table 1 describes datasets that

---

[1]https://github.com/anonymous/anonymous

were used in Chen et al. (2019) as benchmarks for their experiments, already preprocessed and split into training and test sets, and published publicly in a very convenient way on github[2]. The bottom section of Table 1 describes more datasets that we added, mainly small binary-classification datasets, to have more variety. The datasets vary by the number of samples, features, and classes.

| Dataset name | #Samples | #Featues | #Classes | References |
|---|---|---|---|---|
| breast-cancer | 546 | 10 | 2 | Chen et al. (2019) |
| covtype | 400000 | 54 | 7 | Chen et al. (2019) |
| cod-rna | 59535 | 8 | 2 | Chen et al. (2019) |
| diabetes | 614 | 8 | 2 | Chen et al. (2019) |
| Fashion-MNIST | 60000 | 784 | 10 | Chen et al. (2019) |
| ijcnn1 | 49990 | 22 | 2 | Chen et al. (2019) |
| MNIST | 60000 | 784 | 10 | Chen et al. (2019) |
| sensorless | 48509 | 48 | 11 | Chen et al. (2019) |
| webspam | 300000 | 254 | 2 | Chen et al. (2019) |
| MNIST 2 vs. 6 | 11876 | 784 | 2 | Chen et al. (2019) |
| electricity | 45312 | 8 | 2 | https://www.openml.org/d/151 |
| drybean | 13611 | 16 | 7 | Koklu & Ozkan (2020); mis (2020) |
| adult | 32561 | 14(*) | 2 | mis (1996) |
| banknote | 1372 | 4 | 2 | mis (2013) |
| gender-by-voice | 3168 | 20 | 2 | https://www.openml.org/d/43437 |
| waveform | 5000 | 40 | 2 | https://www.openml.org/d/979 |
| wind | 6574 | 14 | 2 | https://www.openml.org/d/847 |
| speech | 3686 | 400 | 2 | https://www.openml.org/d/40910 |

Table 1: Datasets used to evaluate our method. (*) - dataset contained categorical features, which were preprocessed with label encoding.

## 5.3 EVALUATED ADVERSARIAL ATTACKS

We evaluated our method with untargeted attacks, with 2 different norms - $L_2$ and $L_\infty$. For the attack methods, we used four black-box attacks which are relevant to tree-based models: Sign-Opt attack (Cheng et al., 2019) , OPT attack (Cheng et al., 2018) , HopSkipJump attack (Chen et al., 2020) and Cube attack (Andriushchenko & Hein, 2019). We used one white-box attack specifically for trees-based models: Leaf-Tuple attack (Zhang et al., 2020). To execute our attacks we used implementation published by Zhang et al. (2020) on github[3].

## 5.4 EXPERIMENTS RESULTS

We applied our method and calculated ROC-AUC and PR-AUC for each combination of the norm, attack method, dataset, and model algorithm. We calculated the difference between our method's performance, the OC-score method's performance, and the performance of a detector trained on the original representation. All of the distributions of the differences are shown in Appendix A as boxplots. Due to space constraints, we only show here plots comparing our method to OC-score for $L_2$ norm, in Figure 2, for experiments targeting XGBoost tree ensembles and in Figure 3, for experiments that targeting RandomForest tree ensembles. Each row is a different attack method, each white point is an experiment on a specific dataset, and the red vertical line is a total mean of all the experiments together.

The figures show the spread of the differences in ROC-AUC and PR-AUC for each norm and attack combination. The boxplot shows us the different quartiles and the median. As we can see in the figures, the total mean and the median of each section is positive, which means that our metrics yielded better results for the new representation in most of our experiments.

The full raw metrics for each one of the experiments are shown in tables in Appendix B for XGBoost and in appendix C for RandomForest.

---

[2]https://github.com/chenhongge/RobustTrees/blob/master/data/download_data.sh

[3]https://github.com/chong-z/tree-ensemble-attack

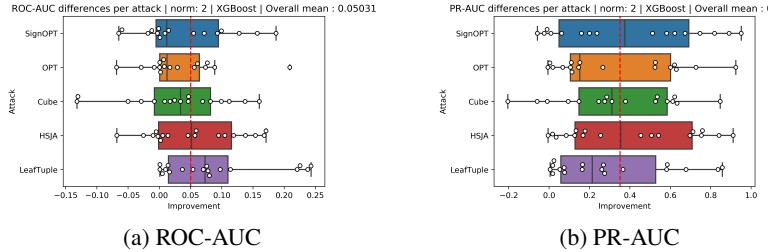

(a) ROC-AUC

(b) PR-AUC

Figure 2: XGBoost experiments metrics differences between the new method and OC-score for $L_2$ norm.

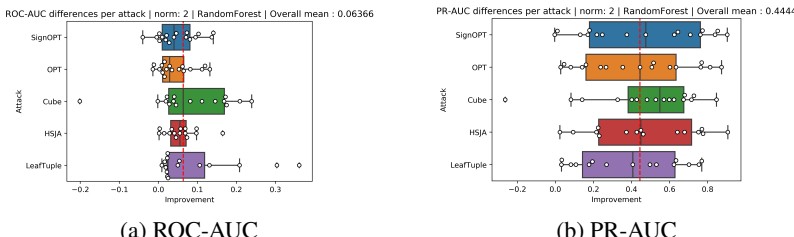

(a) ROC-AUC

(b) PR-AUC

Figure 3: RandomForest experiments metrics differences between the new method and OC-score for $L_2$ norm.

Of our 338 experiments, our method had the best performance in 107 of them and was tied with one of the other methods as the best in 134 other experiments, which means our method was successful in 71.73% of the experiments. We used the Friedman test on the ROC-AUC metric to validate the statistical significance of differences between the evaluated methods and datasets (Demšar, 2006). The above test is a non-parametric test that does not assume anything about the results distributions and is used when dealing with multiple methods and multiple datasets. The null hypothesis that the 3 methods perform the same was rejected with $F_F(338, 3) = 288$ with $p < 0.01$. As a second step, we used Nemneyi post-hoc test, which is often used as a second step if it is possible to reject the null hypothesis with a Friedman test and test for superiority between the different methods, and concluded that our new representation method outperforms the OC-score method and the usage of a machine learning detector based on the original representations with $p < 0.01$.

## 6 DISCUSSION

As we showed in our experiments, using representation learning to learn a better version of the dataset given a trained model helps to extract better performance when trying to protect our model against adversarial evasion attacks. We showed this on different datasets, different tree-based ensembles, and with different attack methods.

An important area for discussion is adversarial attacks on tabular data, which are not as intuitive as adversarial attacks on more continuous inputs such as images or audio. These usually clip the data to a valid range of values, which is not straightforward for tabular data. This issue is not yet heavily researched but was already approached and discussed by Calzavara et al. (2020) and Vos & Verwer (2021), where for each dataset, they defined a set of rewriting rules and possible budget with which to change the features. We chose not to take up that challenge in this work, but for full context, we added in Appendix D statistics regarding our generated adversarial samples.

While analyzing our results, we noticed an interesting outcome of our new representations-generation process. We used UMAP (McInnes et al., 2018) to extract 2-dimensional versions of the original representations and our new ones. As we can see in Figure 4 on the left side, the reduction of the original representations, there the adversarial samples are scattered between the normal

samples. When looking at the reduced version of our representations on the right, we see that the adversarial samples gather closely to each other and are separated from most of the rest of the normal samples.

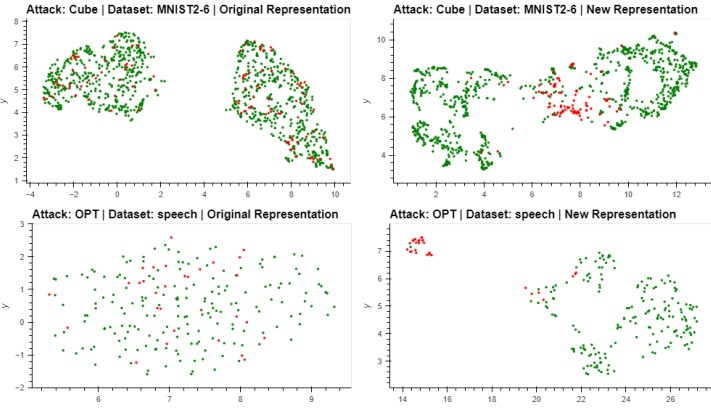

Figure 4: Dimensionality-reduction visualization using UMAP (McInnes et al., 2018). Green points are normal samples, and red points are adversarial samples. On the left, we can see a reduction of the original representations, and on the right, we can see a reduction of the new embedded representations.

## 7  CONCLUSIONS AND FUTURE WORK

In this work, we presented a method to train an adversarial samples detector based on new representations of a dataset based on the target model trained on it. We showed that using our new representation shows overall improvement compared to the current state-of-the-art detection of adversarial samples on tree models without changing the original model internals and compared to using the original samples representations to train a detector classifier.

As for future work, our new method should also be tested on robust versions of tree ensembles to see if it affects the results. Another direction is to create a unified detector for all attacks together. Our detector method was based on generating adversarial samples for each attack method and training a separate model. However, the normal data behave the same regardless of the attack method, and a new unified approach should be researched.

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

## A APPENDIX A - FULL METRICS DIFFERENCES FIGURES

Here are all the different figures comparing our method to OC-Score and a detector based on the original representations of the dataset. For the experiments compared to the OC-score (Figures 5, 6, 9, and 10) the total mean and the median of each section is positive in all of them, which means that our metrics yielded better results for the new representation in most of our experiments. When looking at figures that compare our new method to the original representations (Figures 7, 8, 11, and 12) we can see the ROC-AUC and PR-AUC overall mean of the experiments is positive beside the experiments targeted XGBoost with $L_\infty$ norm which the overall mean is negative.

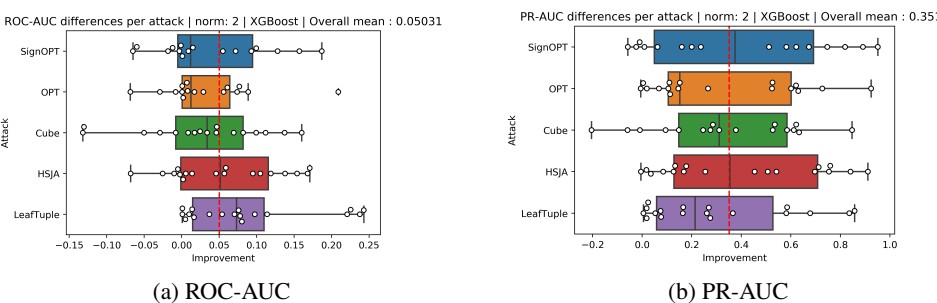

(a) ROC-AUC          (b) PR-AUC

Figure 5: XGBoost experiments metrics differences between the new method and OC-score for $L_2$ norm.

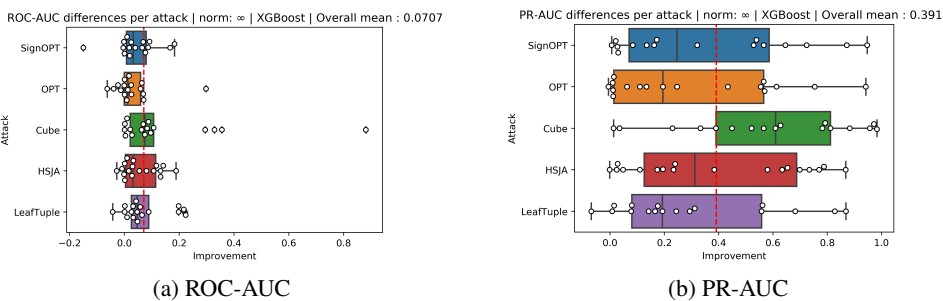

(a) ROC-AUC          (b) PR-AUC

Figure 6: XGBoost experiments metrics differences between the new method and OC-score for $L_\infty$ norm.

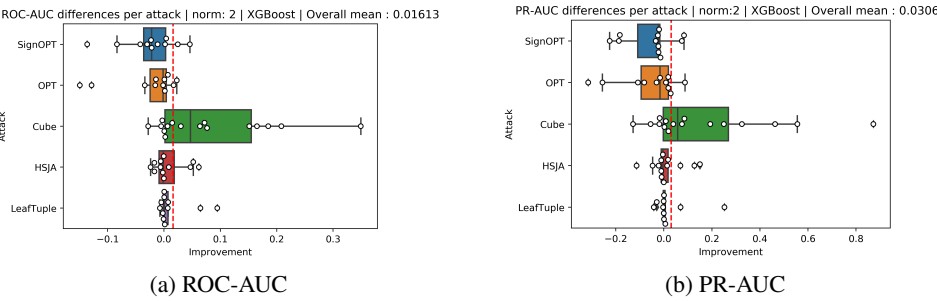

(a) ROC-AUC

(b) PR-AUC

Figure 7: XGBoost experiments metrics differences between the new method and original representation for $L_2$ norm.

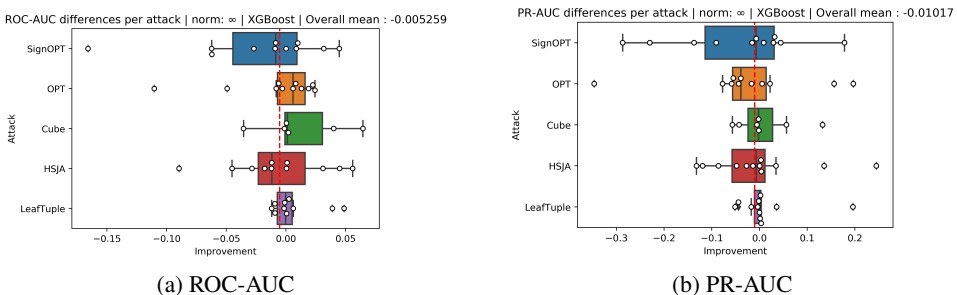

(a) ROC-AUC

(b) PR-AUC

Figure 8: XGBoost experiments metrics differences between the new method and original representation for $L_\infty$ norm.

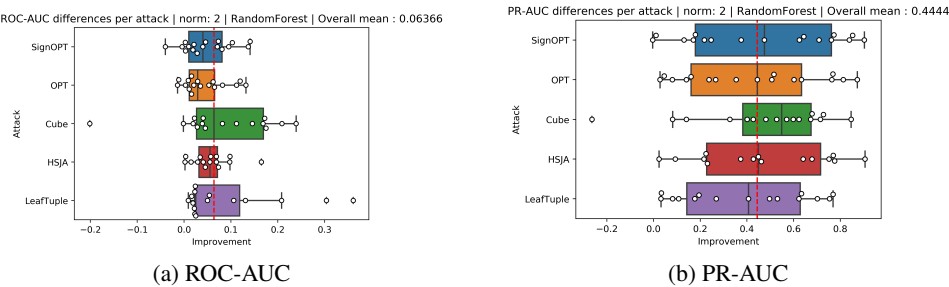

(a) ROC-AUC

(b) PR-AUC

Figure 9: RandomForest experiments metrics differences between the new method and OC-score for $L_2$ norm.

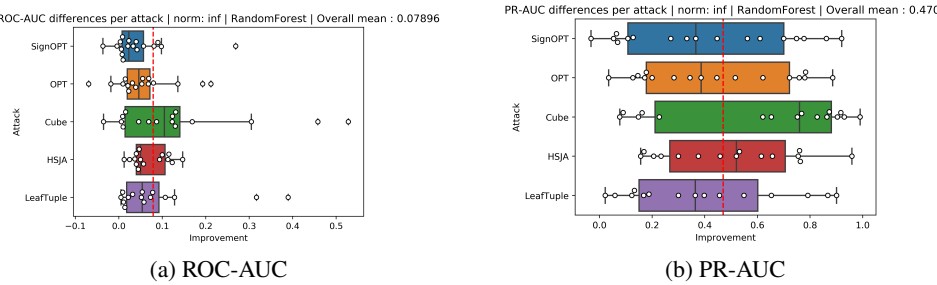

(a) ROC-AUC  (b) PR-AUC

Figure 10: RandomForest experiments metrics differences between the new method and OC-score for $L_\infty$ norm.

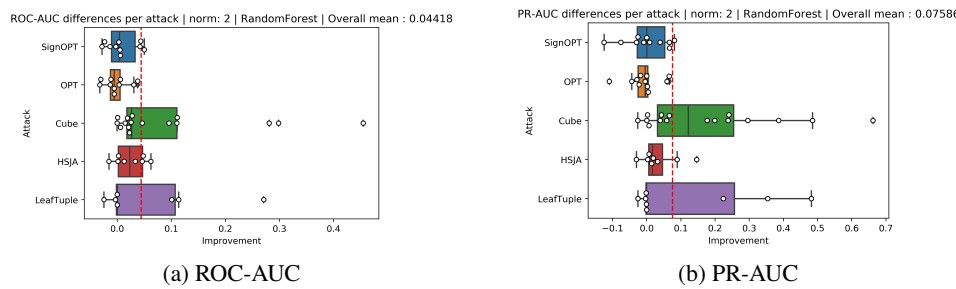

(a) ROC-AUC  (b) PR-AUC

Figure 11: RandomForest experiments metrics differences between the new method and original representation for $L_2$ norm.

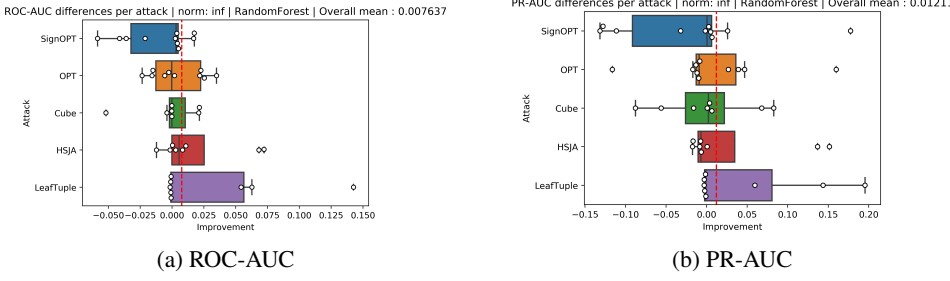

(a) ROC-AUC  (b) PR-AUC

Figure 12: RandomForest experiments metrics differences between the new method and original representation for $L_\infty$ norm.

# B  APPENDIX B - XGBOOST FULL EXPERIMENTS RESULTS

List of tables with the raw experiments metrics for each dataset, attack method, and norm for the XGBoost target experiments. In bold is the method or methods that achieved the highest value. Rows that fill in hyphens are cases where the adversarial sample creation process failed.

# C  APPENDIX C - RANDOMFOREST FULL EXPERIMENTS RESULTS

List of tables with the raw experiments metrics for each dataset, attack method, and norm for the RandomForest target experiments. In bold is the method or methods that achieved the highest value. Rows that fill in hyphens are cases where the adversarial sample creation process failed.

| | Sign-OPT $L_2$ | | | | | | Sign-OPT $L_\infty$ | | | | | |
|---|---|---|---|---|---|---|---|---|---|---|---|---|
| | PRC-AUC | | | ROC-AUC | | | PRC-AUC | | | ROC-AUC | | |
| Dataset | New | Original | OC-score | New | Original | OC-score | New | Original | OC-score | New | Original | OC-score |
| breast-cancer | 0.988 | **0.9919** | 0.958 | 0.987 | **0.9919** | 0.965 | **0.9971** | 0.997 | 0.960 | **0.997** | **0.997** | 0.970 |
| covtype | **1.0** | **1.0** | 0.049 | **1.0** | **1.0** | 0.843 | **1.0** | **1.0** | 0.053 | **1.0** | **1.0** | 0.833 |
| cod-rna | 0.743 | **0.7655** | 0.161 | **0.9795** | 0.955 | 0.880 | **0.7773** | 0.599 | 0.238 | **0.9808** | 0.936 | 0.889 |
| diabetes | 0.672 | **0.855** | 0.730 | 0.707 | **0.8432** | 0.772 | 0.580 | **0.8668** | 0.445 | 0.702 | **0.8678** | 0.615 |
| Fashion-MNIST | **1.0** | **1.0** | 0.157 | **1.0** | **1.0** | 0.839 | **0.9999** | **0.9999** | 0.237 | **1.0** | **1.0** | 0.908 |
| ijcnn1 | **1.0** | **1.0** | 0.181 | **1.0** | **1.0** | 0.907 | **1.0** | **1.0** | 0.276 | **1.0** | **1.0** | 0.935 |
| MNIST | **1.0** | **1.0** | 0.327 | **1.0** | **1.0** | 0.945 | **1.0** | **1.0** | 0.434 | **1.0** | **1.0** | 0.960 |
| MNIST2-6 | **0.9999** | **0.9999** | 0.994 | **1.0** | **1.0** | 0.999 | **1.0** | **1.0** | 0.991 | **1.0** | **1.0** | 0.999 |
| Sensorless | 0.887 | 0.803 | **0.9111** | 0.980 | 0.934 | **0.9982** | **0.9709** | 0.939 | 0.887 | 0.997 | 0.987 | **0.9984** |
| webspam | **1.0** | **1.0** | 0.253 | **1.0** | **1.0** | 0.985 | **1.0** | **1.0** | 0.354 | **1.0** | **1.0** | 0.990 |
| electricity | 0.751 | **0.9763** | 0.129 | 0.955 | **0.997** | 0.883 | 0.829 | **0.9661** | 0.300 | 0.970 | **0.9968** | 0.901 |
| drybean | 0.942 | **0.9617** | 0.705 | 0.975 | **0.9862** | 0.974 | **0.9585** | 0.914 | 0.796 | **0.9868** | 0.955 | 0.967 |
| adult | **1.0** | **1.0** | 0.109 | **1.0** | **1.0** | 0.813 | **1.0** | **1.0** | 0.128 | **1.0** | **1.0** | 0.817 |
| banknote | 0.968 | **0.99** | 0.768 | 0.970 | **0.9918** | 0.961 | **0.9699** | 0.961 | 0.798 | **0.9822** | 0.982 | 0.957 |
| gender-by-voice | **0.9854** | 0.983 | 0.887 | **0.9897** | 0.989 | 0.978 | 0.982 | **0.9964** | 0.960 | 0.990 | **0.9975** | 0.993 |
| waveform | 0.540 | **0.5538** | 0.380 | 0.803 | 0.801 | **0.8628** | 0.467 | **0.5575** | 0.461 | 0.717 | 0.779 | **0.8665** |
| wind | 0.695 | **0.8821** | 0.183 | 0.858 | **0.9414** | 0.730 | 0.526 | **0.7558** | 0.205 | 0.819 | **0.8812** | 0.742 |
| speech | 0.990 | 0.915 | **1.0** | 0.999 | 0.995 | **1.0** | **0.9943** | 0.965 | 0.967 | **0.9987** | 0.990 | 0.998 |

Table 2: Sign-OPT XGBoost Experiments Results

| | OPT $L_2$ | | | | | | OPT $L_\infty$ | | | | | |
|---|---|---|---|---|---|---|---|---|---|---|---|---|
| | PRC-AUC | | | ROC-AUC | | | PRC-AUC | | | ROC-AUC | | |
| Dataset | New | Original | OC-score | New | Original | OC-score | New | Original | OC-score | New | Original | OC-score |
| breast-cancer | 0.992 | **0.9956** | 0.867 | 0.992 | **0.9952** | 0.925 | 0.973 | **0.9737** | 0.920 | **0.9788** | 0.978 | 0.968 |
| covtype | **1.0** | **1.0** | 0.056 | **1.0** | **1.0** | 0.768 | **1.0** | **1.0** | 0.072 | **1.0** | **1.0** | 0.856 |
| cod-rna | **0.804** | 0.716 | 0.183 | **0.9808** | 0.958 | 0.892 | **0.8035** | 0.647 | 0.234 | **0.9774** | 0.953 | 0.907 |
| diabetes | 0.652 | **0.8434** | 0.594 | 0.675 | **0.8612** | 0.741 | **0.8455** | 0.751 | 0.699 | **0.8174** | 0.745 | 0.736 |
| Fashion-MNIST | **1.0** | **1.0** | 0.370 | **1.0** | **1.0** | 0.923 | **1.0** | **1.0** | 0.445 | **1.0** | **1.0** | 0.936 |
| ijcnn1 | **1.0** | **1.0** | 0.273 | **1.0** | **1.0** | 0.926 | **1.0** | **1.0** | 0.246 | **1.0** | **1.0** | 0.941 |
| MNIST | **1.0** | **1.0** | 0.476 | **1.0** | **1.0** | 0.971 | **1.0** | **1.0** | 0.566 | **1.0** | **1.0** | 0.975 |
| MNIST2-6 | **1.0** | **1.0** | 0.996 | **1.0** | **1.0** | **1.0** | **1.0** | **1.0** | 0.986 | **1.0** | **1.0** | 0.999 |
| sensorless | **0.9929** | 0.974 | 0.883 | **0.9998** | 0.999 | 0.999 | **0.9871** | 0.965 | 0.877 | **0.999** | 0.993 | 0.998 |
| webspam | **1.0** | **1.0** | 0.399 | **1.0** | **1.0** | 0.992 | **1.0** | **1.0** | 0.387 | **1.0** | **1.0** | 0.991 |
| electricity | 0.774 | **0.8818** | 0.248 | 0.948 | **0.982** | 0.887 | 0.852 | **0.891** | 0.286 | **0.9731** | 0.960 | 0.904 |
| drybean | **0.9536** | 0.935 | 0.848 | **0.9861** | 0.969 | 0.984 | **0.9396** | 0.934 | 0.877 | 0.964 | 0.970 | **0.9869** |
| adult | **1.0** | **1.0** | 0.076 | **1.0** | **1.0** | 0.791 | **1.0** | **1.0** | 0.058 | **1.0** | **1.0** | 0.702 |
| banknote | **0.9893** | 0.950 | 0.911 | **0.9906** | 0.984 | 0.976 | **0.956** | 0.942 | 0.876 | 0.974 | 0.967 | **0.9767** |
| gender-by-voice | 0.964 | **0.9943** | 0.812 | 0.981 | **0.9964** | 0.964 | 0.976 | **0.9926** | 0.842 | 0.987 | **0.9951** | 0.974 |
| waveform | 0.538 | **0.6195** | 0.392 | 0.784 | 0.777 | **0.8524** | 0.590 | **0.6673** | 0.395 | 0.819 | 0.800 | **0.8813** |
| wind | 0.476 | **0.7908** | 0.210 | 0.777 | **0.9054** | 0.785 | 0.423 | **0.7694** | 0.176 | 0.755 | **0.8653** | 0.729 |
| speech | **0.9917** | 0.983 | 0.973 | **0.9986** | 0.997 | 0.998 | **0.9957** | 0.799 | 0.984 | 0.9996 | 0.977 | **1.0** |

Table 3: OPT XGBoost Experiments Results

| | HSJA $L_2$ | | | | | | HSJA $L_\infty$ | | | | | |
|---|---|---|---|---|---|---|---|---|---|---|---|---|
| | PRC-AUC | | | ROC-AUC | | | PRC-AUC | | | ROC-AUC | | |
| Dataset | New | Original | OC-score | New | Original | OC-score | New | Original | OC-score | New | Original | OC-score |
| breast-cancer | **0.9909** | 0.988 | 0.799 | **0.9905** | 0.988 | 0.923 | 0.982 | **0.9956** | 0.934 | 0.987 | **0.9958** | 0.954 |
| covtype | **1.0** | **1.0** | 0.159 | **1.0** | **1.0** | 0.832 | **1.0** | **1.0** | 0.230 | **1.0** | **1.0** | 0.811 |
| cod-rna | **0.8193** | 0.669 | 0.107 | **0.9879** | 0.926 | 0.817 | **0.8389** | 0.594 | 0.185 | **0.9891** | 0.944 | 0.847 |
| diabetes | **0.7947** | 0.786 | 0.522 | **0.834** | 0.830 | 0.708 | 0.787 | **0.8219** | 0.516 | 0.790 | **0.8325** | 0.710 |
| Fashion-MNIST | 0.994 | **0.998** | 0.235 | **1.0** | **1.0** | 0.881 | **1.0** | 0.9997 | 0.266 | **1.0** | **1.0** | 0.868 |
| ijcnn1 | **1.0** | **1.0** | 0.089 | **1.0** | **1.0** | 0.846 | **1.0** | **1.0** | 0.132 | **1.0** | **1.0** | 0.893 |
| MNIST | **1.0** | **1.0** | 0.493 | **1.0** | **1.0** | 0.954 | **1.0** | **1.0** | 0.420 | **1.0** | **1.0** | 0.945 |
| MNIST2-6 | **1.0** | **1.0** | 0.965 | **1.0** | **1.0** | 0.998 | **1.0** | **1.0** | 0.972 | **1.0** | **1.0** | 0.998 |
| sensorless | **0.8562** | 0.729 | 0.839 | 0.972 | 0.920 | **0.9973** | **0.9175** | 0.782 | 0.893 | 0.990 | 0.934 | **0.9984** |
| webspam | **0.9859** | 0.986 | 0.232 | 0.9999 | **1.0** | 0.986 | **1.0** | **1.0** | 0.212 | **1.0** | **1.0** | 0.984 |
| electricity | 0.795 | **0.9084** | 0.098 | 0.962 | **0.9789** | 0.824 | **0.8877** | 0.853 | 0.189 | **0.9841** | 0.953 | 0.852 |
| drybean | **0.9568** | 0.943 | 0.872 | **0.9826** | 0.974 | 0.977 | **0.9561** | 0.953 | 0.846 | **0.9795** | 0.979 | 0.978 |
| adult | **1.0** | **1.0** | 0.458 | **1.0** | **1.0** | 0.905 | **1.0** | **1.0** | 0.366 | **1.0** | **1.0** | 0.883 |
| banknote | **0.8879** | 0.851 | 0.742 | 0.889 | **0.9017** | 0.901 | 0.895 | **0.901** | 0.658 | **0.9265** | 0.907 | 0.869 |
| gender-by-voice | 0.920 | **0.9668** | 0.787 | 0.952 | **0.9758** | 0.962 | 0.961 | **0.9882** | 0.721 | 0.975 | **0.9927** | 0.947 |
| waveform | **0.6012** | 0.592 | 0.493 | 0.774 | 0.768 | **0.883** | 0.680 | **0.692** | 0.372 | **0.8323** | 0.815 | 0.831 |
| wind | 0.763 | **0.7846** | 0.309 | 0.881 | **0.8814** | 0.824 | 0.712 | **0.8311** | 0.328 | 0.870 | **0.9151** | 0.789 |
| speech | **0.9965** | 0.994 | 0.989 | **0.9993** | 0.999 | 0.999 | 0.996 | 0.994 | **0.9995** | 0.999 | 0.998 | **1.0** |

Table 4: HSJA XGBoost Experiments Results

| | Cube $L_2$ | | | | | | Cube $L_\infty$ | | | | | |
|---|---|---|---|---|---|---|---|---|---|---|---|---|
| | PRC-AUC | | | ROC-AUC | | | PRC-AUC | | | ROC-AUC | | |
| Dataset | New | Original | OC-score | New | Original | OC-score | New | Original | OC-score | New | Original | OC-score |
| breast-cancer | 0.537 | 0.212 | **0.7413** | 0.756 | 0.591 | **0.8878** | 0.869 | **0.9257** | 0.303 | 0.931 | **0.9665** | 0.575 |
| covtype | **0.9823** | 0.020 | 0.045 | **0.9998** | 0.649 | 0.838 | **1.0** | **1.0** | 0.043 | **1.0** | **1.0** | 0.893 |
| cod-rna | **0.7678** | 0.212 | 0.183 | **0.985** | 0.800 | 0.873 | **0.8643** | 0.732 | 0.343 | **0.9841** | 0.944 | 0.915 |
| diabetes | **0.5437** | 0.458 | 0.258 | **0.7935** | 0.717 | 0.724 | 0.567 | **0.6097** | 0.532 | **0.7316** | 0.667 | 0.710 |
| Fashion-MNIST | **0.9094** | 0.538 | 0.098 | **0.9951** | 0.798 | 0.850 | **0.8929** | 0.750 | 0.014 | **0.9996** | 0.999 | 0.903 |
| ijcnn1 | **0.773** | 0.310 | 0.246 | **0.987** | 0.915 | 0.905 | 0.998 | **0.9998** | 0.216 | **1.0** | **1.0** | 0.912 |
| MNIST | **0.7755** | 0.263 | 0.202 | **0.9846** | 0.735 | 0.926 | **1.0** | **1.0** | 0.028 | **1.0** | **1.0** | 0.917 |
| MNIST2-6 | 0.968 | **0.9853** | 0.977 | 0.990 | 0.995 | **0.9979** | **1.0** | **1.0** | 0.987 | **1.0** | **1.0** | 0.999 |
| sensorless | **0.8793** | 0.650 | 0.595 | 0.994 | 0.916 | **0.9961** | **1.0** | **1.0** | 0.770 | **1.0** | **1.0** | 0.996 |
| webspam | **0.9185** | 0.843 | 0.297 | **0.9987** | 0.984 | 0.990 | **0.9994** | **0.9994** | 0.390 | **1.0** | **1.0** | 0.990 |
| electricity | **0.7746** | 0.671 | 0.312 | **0.9666** | 0.887 | 0.904 | **0.9931** | 0.986 | 0.652 | **0.9996** | 0.995 | 0.969 |
| drybean | **0.9954** | 0.987 | 0.720 | **0.9996** | 0.999 | 0.975 | **1.0** | **1.0** | 0.610 | **1.0** | **1.0** | 0.937 |
| adult | 0.944 | **0.9454** | 0.315 | **0.993** | 0.985 | 0.941 | **1.0** | **1.0** | 0.551 | **1.0** | **1.0** | 0.998 |
| banknote | **0.6201** | 0.538 | 0.469 | 0.971 | 0.862 | **0.9893** | **0.9973** | **0.9973** | 0.364 | **0.9992** | **0.9992** | 0.923 |
| gender-by-voice | 0.811 | **0.866** | 0.663 | 0.915 | 0.943 | **0.9439** | **1.0** | **1.0** | 0.117 | **1.0** | **1.0** | 0.671 |
| waveform | **0.9614** | 0.959 | 0.426 | **0.9754** | 0.969 | 0.838 | **1.0** | **1.0** | 0.188 | **1.0** | **1.0** | 0.705 |
| wind | **1.0** | **1.0** | 0.754 | **1.0** | **1.0** | 0.966 | **1.0** | **1.0** | 0.017 | **1.0** | **1.0** | 0.119 |
| speech | - | - | - | - | - | - | - | - | - | - | - | - |

Table 5: Cube XGBoost Experiments Results

| | Leaf-Tuple $L_2$ | | | | | | Leaf-Tuple $L_\infty$ | | | | | |
|---|---|---|---|---|---|---|---|---|---|---|---|---|
| | PRC-AUC | | | ROC-AUC | | | PRC-AUC | | | ROC-AUC | | |
| Dataset | New | Original | OC-score | New | Original | OC-score | New | Original | OC-score | New | Original | OC-score |
| breast-cancer | **0.9954** | 0.989 | 0.921 | **0.9954** | 0.988 | 0.915 | 0.871 | 0.835 | **0.9399** | 0.893 | 0.854 | **0.9356** |
| covtype | **0.9962** | 0.723 | 0.118 | **0.9995** | 0.934 | 0.761 | **0.9969** | 0.783 | 0.110 | **0.9999** | 0.950 | 0.782 |
| cod-rna | 0.932 | **0.9477** | 0.540 | 0.981 | **0.9822** | 0.861 | 0.951 | **0.97** | 0.614 | 0.986 | **0.9891** | 0.891 |
| diabetes | **0.7227** | 0.653 | 0.557 | **0.7826** | 0.688 | 0.685 | 0.561 | **0.5782** | 0.419 | 0.643 | **0.6522** | 0.611 |
| Fashion-MNIST | **0.9627** | 0.948 | 0.367 | **0.9891** | 0.973 | 0.759 | **0.9544** | 0.951 | 0.393 | **0.9868** | 0.978 | 0.786 |
| ijcnn1 | 0.946 | **0.9584** | 0.668 | 0.984 | **0.9867** | 0.906 | 0.960 | **0.9711** | 0.744 | 0.988 | **0.9899** | 0.935 |
| MNIST | 0.995 | **0.9953** | 0.410 | **0.9982** | 0.997 | 0.773 | **0.9972** | 0.997 | 0.435 | **0.9993** | 0.999 | 0.800 |
| MNIST2-6 | 0.999 | **0.9997** | 0.981 | 0.999 | **0.9997** | 0.989 | 0.999 | **0.9995** | 0.983 | 0.999 | **0.9995** | 0.988 |
| Sensorless | **0.9657** | 0.965 | 0.889 | **0.9855** | 0.985 | 0.969 | **0.9712** | 0.969 | 0.891 | **0.9886** | 0.986 | 0.956 |
| webspam | **1.0** | **1.0** | 0.322 | **1.0** | **1.0** | 0.946 | **1.0** | **1.0** | 0.318 | **1.0** | **1.0** | 0.942 |
| electricity | 0.987 | **0.987** | 0.712 | **0.9955** | 0.995 | 0.925 | **0.9935** | 0.992 | 0.700 | **0.9976** | 0.997 | 0.933 |
| drybean | **1.0** | **1.0** | 0.836 | **1.0** | **1.0** | 0.963 | **1.0** | **1.0** | 0.807 | **1.0** | **1.0** | 0.952 |
| adult | **1.0** | **1.0** | 0.982 | **1.0** | **1.0** | 0.999 | **1.0** | **1.0** | **1.0** | **1.0** | **1.0** | **1.0** |
| banknote | 0.997 | **1.0** | 0.728 | 0.998 | **1.0** | 0.920 | **1.0** | **1.0** | 0.756 | **1.0** | **1.0** | 0.953 |
| gender-by-voice | **1.0** | **1.0** | 0.947 | **1.0** | **1.0** | 0.986 | **1.0** | **1.0** | 0.922 | **1.0** | **1.0** | 0.974 |
| waveform | **1.0** | **1.0** | 0.164 | **1.0** | **1.0** | 0.757 | **1.0** | **1.0** | 0.172 | **1.0** | **1.0** | 0.776 |
| wind | **1.0** | **1.0** | 0.995 | **1.0** | **1.0** | 0.999 | **1.0** | **1.0** | 0.834 | **1.0** | **1.0** | 0.999 |
| speech | **1.0** | **1.0** | 0.976 | **1.0** | **1.0** | 0.995 | **1.0** | 0.9999 | 0.992 | **0.9999** | **0.9999** | 0.998 |

Table 6: Leaf-Tuple XGBoost Experiments Results

| | Sign-OPT $L_2$ | | | | | | Sign-OPT $L_\infty$ | | | | | |
|---|---|---|---|---|---|---|---|---|---|---|---|---|
| | PRC-AUC | | | ROC-AUC | | | PRC-AUC | | | ROC-AUC | | |
| Dataset | New | Original | OC-score | New | Original | OC-score | New | Original | OC-score | New | Original | OC-score |
| breast-cancer | 0.992 | 0.983 | **0.9952** | 0.992 | 0.986 | **0.9962** | 0.965 | **0.997** | 0.997 | 0.961 | 0.997 | **0.9974** |
| covtype | **1.0** | **1.0** | 0.097 | **1.0** | **1.0** | 0.859 | **1.0** | **1.0** | 0.080 | **1.0** | **1.0** | 0.731 |
| cod-rna | **0.9572** | 0.918 | 0.248 | **0.9982** | 0.995 | 0.925 | **0.9266** | 0.749 | 0.178 | **0.996** | 0.979 | 0.898 |
| diabetes | 0.840 | **0.8485** | 0.593 | **0.8593** | 0.854 | 0.764 | 0.738 | **0.8493** | 0.669 | 0.815 | **0.8735** | 0.809 |
| Fashion-MNIST | **1.0** | **1.0** | 0.147 | **1.0** | **1.0** | 0.919 | **1.0** | **1.0** | 0.223 | **1.0** | **1.0** | 0.943 |
| ijcnn1 | **1.0** | **1.0** | 0.375 | 1.000 | **1.0** | 0.972 | **1.0** | **1.0** | 0.390 | **1.0** | **1.0** | 0.958 |
| MNIST | **1.0** | **1.0** | 0.238 | **1.0** | **1.0** | 0.929 | **1.0** | **1.0** | 0.553 | **1.0** | **1.0** | 0.967 |
| MNIST2-6 | **1.0** | **1.0** | 0.869 | **1.0** | **1.0** | 0.990 | **1.0** | **1.0** | 0.872 | **1.0** | **1.0** | 0.990 |
| sensorless | **0.8165** | 0.735 | 0.637 | **0.9842** | 0.942 | 0.981 | **0.973** | 0.968 | 0.641 | **0.9982** | 0.994 | 0.975 |
| webspam | **1.0** | **1.0** | 0.161 | **1.0** | **1.0** | 0.981 | **1.0** | **1.0** | 0.138 | **1.0** | **1.0** | 0.979 |
| electricity | 0.939 | **0.9644** | 0.295 | 0.995 | **0.9975** | 0.892 | 0.968 | **0.9696** | 0.406 | **0.9981** | 0.993 | 0.917 |
| drybean | **0.9808** | 0.914 | 0.810 | **0.9897** | 0.946 | 0.987 | **0.9739** | 0.948 | 0.866 | **0.9945** | 0.977 | 0.991 |
| adult | **1.0** | 0.9998 | 0.227 | **1.0** | **1.0** | 0.863 | **1.0** | **1.0** | 0.300 | **1.0** | **1.0** | 0.910 |
| banknote | 0.886 | **0.9614** | 0.875 | 0.941 | 0.969 | **0.9809** | **0.9847** | 0.982 | 0.920 | **0.9888** | 0.985 | 0.981 |
| gender-by-voice | 0.969 | **0.9988** | 0.751 | 0.986 | **0.9992** | 0.946 | **0.9957** | 0.989 | 0.942 | **0.9976** | 0.995 | 0.989 |
| waveform | **0.8227** | 0.756 | 0.447 | **0.9182** | 0.868 | 0.897 | 0.787 | **0.9148** | 0.516 | 0.914 | **0.9552** | 0.918 |
| wind | 0.784 | **0.9087** | 0.309 | 0.943 | **0.9661** | 0.897 | 0.745 | **0.8765** | 0.379 | 0.934 | **0.9551** | 0.894 |
| speech | - | - | - | - | - | - | - | - | - | - | - | - |

Table 7: Sign-OPT RandomForest Experiments Results

| | OPT $L_2$ | | | | | | OPT $L_\infty$ | | | | | |
|---|---|---|---|---|---|---|---|---|---|---|---|---|
| | PRC-AUC | | | ROC-AUC | | | PRC-AUC | | | ROC-AUC | | |
| Dataset | New | Original | OC-score | New | Original | OC-score | New | Original | OC-score | New | Original | OC-score |
| breast-cancer | 0.975 | **0.9926** | 0.947 | 0.982 | **0.9935** | 0.973 | 0.980 | **0.9932** | 0.834 | 0.978 | **0.9937** | 0.909 |
| covtype | **1.0** | **1.0** | 0.127 | **1.0** | **1.0** | 0.868 | **1.0** | **1.0** | 0.114 | **1.0** | **1.0** | 0.807 |
| cod-rna | **0.9316** | 0.927 | 0.163 | **0.9977** | 0.994 | 0.878 | **0.9069** | 0.747 | 0.148 | **0.9948** | 0.973 | 0.915 |
| diabetes | 0.747 | **0.7514** | 0.509 | 0.784 | **0.8161** | 0.719 | **0.8889** | 0.842 | 0.545 | **0.913** | 0.878 | 0.701 |
| Fashion-MNIST | **1.0** | **1.0** | 0.236 | **1.0** | **1.0** | 0.938 | **1.0** | **1.0** | 0.220 | **1.0** | **1.0** | 0.932 |
| ijcnn1 | **1.0** | **1.0** | 0.366 | **1.0** | **1.0** | 0.947 | **1.0** | **1.0** | 0.379 | **1.0** | **1.0** | 0.945 |
| MNIST | **1.0** | **1.0** | 0.555 | **1.0** | **1.0** | 0.971 | **1.0** | **1.0** | 0.484 | **1.0** | **1.0** | 0.961 |
| MNIST2-6 | **1.0** | 0.9997 | 0.859 | **1.0** | **1.0** | 0.989 | **1.0** | **1.0** | 0.800 | **1.0** | **1.0** | 0.977 |
| Sensorless | **0.9705** | 0.911 | 0.704 | **0.9978** | 0.967 | 0.977 | **0.9772** | 0.938 | 0.694 | **0.9984** | 0.973 | 0.967 |
| webspam | **1.0** | **1.0** | 0.187 | **1.0** | **1.0** | 0.984 | **1.0** | **1.0** | 0.217 | **1.0** | **1.0** | 0.985 |
| electricity | 0.936 | **0.9646** | 0.429 | 0.991 | **0.9964** | 0.956 | 0.975 | **0.992** | 0.529 | 0.997 | **0.9995** | 0.976 |
| drybean | **0.983** | 0.917 | 0.902 | **0.996** | 0.958 | 0.993 | **0.9779** | 0.951 | 0.806 | **0.9937** | 0.971 | 0.984 |
| adult | **1.0** | **1.0** | 0.398 | **1.0** | **1.0** | 0.918 | **1.0** | **1.0** | 0.278 | **1.0** | **1.0** | 0.864 |
| banknote | 0.954 | **0.9975** | 0.908 | 0.968 | **0.9982** | 0.982 | 0.942 | **0.9504** | 0.815 | 0.949 | 0.964 | **0.9672** |
| gender-by-voice | 0.973 | **0.995** | 0.812 | 0.984 | **0.9968** | 0.968 | 0.985 | **0.9947** | 0.950 | 0.991 | **0.9967** | 0.991 |
| waveform | **0.8013** | 0.799 | 0.447 | 0.899 | 0.893 | **0.9103** | 0.659 | **0.6708** | 0.481 | 0.826 | 0.824 | **0.8952** |
| wind | 0.752 | **0.8612** | 0.236 | 0.922 | **0.9273** | 0.810 | 0.740 | **0.8563** | 0.354 | 0.923 | **0.9465** | 0.870 |
| speech | - | - | - | - | - | - | - | - | - | - | - | - |

Table 8: OPT RandomForest Experiments Results

| | HSJA $L_2$ | | | | | | HSJA $L_\infty$ | | | | | |
|---|---|---|---|---|---|---|---|---|---|---|---|---|
| | PRC-AUC | | | ROC-AUC | | | PRC-AUC | | | ROC-AUC | | |
| Dataset | New | Original | OC-score | New | Original | OC-score | New | Original | OC-score | New | Original | OC-score |
| breast-cancer | **0.986** | 0.983 | 0.962 | **0.9875** | 0.986 | 0.984 | 0.980 | **0.9974** | 0.774 | 0.986 | **0.9982** | 0.936 |
| covtype | **1.0** | **1.0** | 0.230 | **1.0** | **1.0** | 0.835 | **1.0** | **1.0** | 0.245 | **1.0** | **1.0** | 0.853 |
| cod-rna | - | - | - | - | - | - | - | - | - | - | - | - |
| diabetes | - | - | - | - | - | - | - | - | - | - | - | - |
| Fashion-MNIST | **0.9999** | **0.9999** | 0.248 | **1.0** | **1.0** | 0.943 | **1.0** | **1.0** | 0.240 | **1.0** | **1.0** | 0.943 |
| ijcnn1 | **1.0** | **1.0** | 0.224 | **1.0** | **1.0** | 0.902 | **1.0** | **1.0** | 0.237 | **1.0** | **1.0** | 0.887 |
| MNIST | **1.0** | **1.0** | 0.321 | **1.0** | **1.0** | 0.931 | **1.0** | **1.0** | 0.385 | **1.0** | **1.0** | 0.953 |
| MNIST2-6 | **1.0** | **1.0** | 0.775 | **1.0** | **1.0** | 0.985 | **1.0** | **1.0** | 0.700 | **1.0** | **1.0** | 0.975 |
| sensorless | **0.9235** | 0.777 | 0.550 | **0.9934** | 0.931 | 0.948 | **0.8885** | 0.737 | 0.430 | **0.9913** | 0.919 | 0.951 |
| webspam | **1.0** | **1.0** | 0.094 | **1.0** | **1.0** | 0.959 | **1.0** | **1.0** | 0.041 | **1.0** | **1.0** | 0.900 |
| electricity | **0.9381** | 0.931 | 0.297 | **0.9935** | 0.991 | 0.920 | 0.947 | **0.9544** | 0.290 | **0.9973** | 0.989 | 0.882 |
| drybean | **0.9676** | 0.948 | 0.874 | **0.991** | 0.978 | 0.989 | **0.9847** | 0.984 | 0.828 | **0.9974** | 0.997 | 0.985 |
| adult | **1.0** | **1.0** | 0.538 | **1.0** | **1.0** | 0.931 | **1.0** | **1.0** | 0.480 | **1.0** | **1.0** | 0.877 |
| banknote | **0.9632** | 0.874 | 0.747 | **0.9642** | 0.916 | 0.909 | 0.943 | **0.9496** | 0.774 | **0.9579** | 0.955 | 0.921 |
| gender-by-voice | 0.955 | **0.985** | 0.724 | 0.973 | **0.9882** | 0.944 | 0.960 | **0.9684** | 0.726 | 0.975 | **0.9765** | 0.930 |
| waveform | **0.736** | 0.704 | 0.308 | **0.8824** | 0.836 | 0.848 | **0.7729** | 0.636 | 0.397 | **0.8893** | 0.821 | 0.849 |
| wind | **0.8744** | 0.860 | 0.425 | **0.9495** | 0.916 | 0.852 | 0.843 | **0.8597** | 0.311 | **0.9519** | 0.941 | 0.858 |
| speech | - | - | - | - | - | - | - | - | - | - | - | - |

Table 9: HSJA RandomForest Experiments Results

| | Cube $L_2$ | | | | | | Cube $L_\infty$ | | | | | |
|---|---|---|---|---|---|---|---|---|---|---|---|---|
| | PRC-AUC | | | ROC-AUC | | | PRC-AUC | | | ROC-AUC | | |
| Dataset | New | Original | OC-score | New | Original | OC-score | New | Original | OC-score | New | Original | OC-score |
| breast-cancer | 0.435 | 0.139 | **0.6983** | 0.738 | 0.283 | **0.9391** | **1.0** | 0.932 | 0.173 | **1.0** | 0.979 | 0.695 |
| covtype | **0.7853** | 0.123 | 0.106 | **0.9911** | 0.710 | 0.752 | **1.0** | **1.0** | 0.070 | **1.0** | **1.0** | 0.831 |
| cod-rna | **0.7844** | 0.299 | 0.111 | **0.9871** | 0.876 | 0.815 | **0.9158** | 0.833 | 0.164 | **0.9955** | 0.974 | 0.872 |
| diabetes | **0.6343** | 0.248 | 0.307 | **0.9185** | 0.620 | 0.750 | 0.384 | **0.4718** | 0.307 | 0.562 | **0.6139** | 0.597 |
| Fashion-MNIST | **0.864** | 0.687 | 0.149 | **0.9939** | 0.898 | 0.912 | **0.9947** | 0.991 | 0.374 | 0.9999 | **1.0** | 0.954 |
| ijcnn1 | **0.8503** | 0.612 | 0.280 | **0.9949** | 0.967 | 0.883 | 0.984 | **1.0** | 0.120 | **1.0** | **1.0** | 0.870 |
| MNIST | **0.9337** | 0.894 | 0.335 | **0.9978** | 0.972 | 0.957 | **1.0** | **1.0** | 0.852 | **1.0** | **1.0** | 0.994 |
| MNIST2-6 | **0.9399** | 0.933 | 0.858 | 0.990 | 0.971 | **0.9914** | **1.0** | **1.0** | 0.910 | **1.0** | **1.0** | 0.991 |
| sensorless | **0.9066** | 0.666 | 0.425 | **0.9924** | 0.882 | 0.947 | **0.9945** | 0.988 | 0.767 | 0.9998 | **1.0** | 0.984 |
| webspam | **0.9345** | 0.735 | 0.311 | **0.9984** | 0.952 | 0.979 | **0.9988** | 0.998 | 0.127 | 0.9999 | **1.0** | 0.913 |
| electricity | **0.9726** | 0.914 | 0.445 | **0.9978** | 0.975 | 0.852 | **0.9988** | 0.998 | 0.095 | 0.9999 | **1.0** | 0.877 |
| drybean | 0.938 | **0.964** | 0.797 | **0.9862** | 0.980 | 0.964 | **1.0** | **1.0** | 0.838 | **1.0** | **1.0** | 0.990 |
| adult | **0.9977** | 0.995 | 0.151 | 0.9999 | | 0.791 | **1.0** | **1.0** | 0.010 | **1.0** | **1.0** | 0.472 |
| banknote | **0.9316** | 0.865 | 0.504 | **0.9853** | 0.964 | 0.957 | 0.944 | **1.0** | 0.176 | 0.996 | **1.0** | 0.927 |
| gender-by-voice | **0.8822** | 0.839 | 0.481 | **0.9799** | 0.965 | 0.941 | **1.0** | **1.0** | 0.347 | **1.0** | **1.0** | 0.869 |
| waveform | 0.954 | **0.9535** | 0.226 | **0.9739** | 0.973 | 0.799 | **1.0** | **1.0** | 0.083 | **1.0** | **1.0** | 0.542 |
| wind | - | - | - | - | - | - | - | - | - | - | - | - |
| speech | - | - | - | - | - | - | - | - | - | - | - | - |

Table 10: Cube RandomForest Experiments Results

| | Leaf-Tuple $L_2$ | | | | | | Leaf-Tuple $L_\infty$ | | | | | |
| | PRC-AUC | | | ROC-AUC | | | PRC-AUC | | | ROC-AUC | | |
| Dataset | New | Original | OC-score | New | Original | OC-score | New | Original | OC-score | New | Original | OC-score |
|---|---|---|---|---|---|---|---|---|---|---|---|---|
| breast-cancer | 0.971 | **0.9964** | 0.939 | 0.972 | **0.9964** | 0.947 | 0.995 | **0.9964** | 0.872 | 0.995 | **0.9965** | 0.922 |
| covtype | - | - | - | - | - | - | - | - | - | - | - | - |
| cod-rna | 0.988 | **0.9895** | 0.581 | **0.997** | 0.997 | 0.891 | 0.993 | **0.9953** | 0.693 | 0.998 | **0.9987** | 0.940 |
| diabetes | - | - | - | - | - | - | - | - | - | - | - | - |
| Fashion-MNIST | **0.8023** | 0.578 | 0.033 | **0.9918** | 0.878 | 0.688 | **0.8884** | 0.693 | 0.021 | **0.9916** | 0.849 | 0.602 |
| ijcnn1 | **0.9939** | 0.994 | 0.912 | **0.9983** | 0.998 | 0.982 | 0.990 | **0.9929** | 0.931 | 0.997 | **0.998** | 0.988 |
| MNIST | **0.7772** | 0.423 | 0.023 | **0.986** | 0.885 | 0.625 | **0.8996** | 0.840 | 0.108 | **0.9883** | 0.934 | 0.860 |
| MNIST2-6 | 0.996 | **0.999** | 0.962 | 0.996 | **0.999** | 0.973 | 0.998 | **0.9989** | 0.976 | 0.998 | **0.9989** | 0.987 |
| Sensorless | **0.6531** | 0.171 | 0.030 | **0.9801** | 0.709 | 0.772 | **0.9427** | 0.799 | 0.042 | **0.9938** | 0.931 | 0.677 |
| webspam | **1.0** | **1.0** | 0.731 | **1.0** | **1.0** | 0.978 | **1.0** | **1.0** | 0.602 | **1.0** | **1.0** | 0.978 |
| electricity | 0.999 | **0.9997** | 0.822 | **1.0** | 0.9999 | 0.976 | 0.996 | **0.999** | 0.828 | 0.999 | **0.9997** | 0.947 |
| drybean | **1.0** | **1.0** | 0.503 | **1.0** | **1.0** | 0.946 | **1.0** | **1.0** | 0.636 | **1.0** | **1.0** | 0.946 |
| adult | **1.0** | **1.0** | 0.469 | **1.0** | **1.0** | 0.991 | **1.0** | **1.0** | 0.812 | **1.0** | **1.0** | 0.995 |
| banknote | **1.0** | **1.0** | 0.805 | **1.0** | **1.0** | 0.977 | **1.0** | **1.0** | 0.544 | **1.0** | **1.0** | 0.893 |
| gender-by-voice | **1.0** | **1.0** | 0.892 | **1.0** | **1.0** | 0.981 | **1.0** | **1.0** | 0.867 | **1.0** | **1.0** | 0.968 |
| waveform | **1.0** | **1.0** | 0.298 | **1.0** | **1.0** | 0.869 | **1.0** | **1.0** | 0.347 | **1.0** | **1.0** | 0.922 |
| wind | **1.0** | **1.0** | 0.366 | **1.0** | **1.0** | 0.950 | **1.0** | **1.0** | 0.451 | **1.0** | **1.0** | 0.986 |
| speech | - | - | - | - | - | - | - | - | - | - | - | - |

Table 11: Leaf-Tuple RandomForest Experiments Results

# D APPENDIX D - ADVERSARIAL SAMPLES INFORMATION

In the following tables, we calculated several statistics about the perturbations generated in our various experiments split by target model type and attack method. Interesting behaviors we noticed:

- We can see that for SignOPT and OPT attacks in all of the experiments, a perturbation was applied to all of the features (max features changed is equal to the mean, which is also equal to the number of features for each dataset).

- The mean perturbation size for datasets adult and drybean seem very high for attack methods HopSkipJumpAttack, Cube, and LeafTuple.

## D.1 XGBOOST PERTURBATION STATISTICS

| | Sign-OPT $L_2$ | | | Sign-OPT $L_\infty$ | | |
|---|---|---|---|---|---|---|
| **Dataset name** | **Max #feature changed** | **Mean #feature changed** | **Mean $\|L_2\|$ perturbation** | **Max #feature changed** | **Mean #feature changed** | **Mean $\|L_\infty\|$ perturbation** |
| breast-cancer | 10 | 10.0 | 0.3148 | 10 | 10.0 | 0.2514 |
| covtype | 54 | 54.0 | 0.058 | 54 | 54.0 | 0.0433 |
| cod-rna | 8 | 8.0 | 0.0405 | 8 | 8.0 | 0.0366 |
| diabetes | 8 | 8.0 | 0.0567 | 8 | 8.0 | 0.0604 |
| Fashion-MNIST | 784 | 784.0 | 0.0514 | 784 | 784.0 | 0.0423 |
| ijcnn1 | 22 | 22.0 | 0.0425 | 22 | 22.0 | 0.0419 |
| MNIST | 784 | 784.0 | 0.06 | 784 | 784.0 | 0.048 |
| Sensorless | 48 | 48.0 | 0.0168 | 48 | 48.0 | 0.0194 |
| webspam | 254 | 254.0 | 0.0051 | 254 | 254.0 | 0.0065 |
| MNIST 2 vs. 6 | 784 | 784.0 | 0.2907 | 784 | 784.0 | 0.2869 |
| electricity | 8 | 8.0 | 0.01 | 8 | 8.0 | 0.0072 |
| drybean | 16 | 16.0 | 0.0176 | 16 | 16.0 | 0.0103 |
| adult | 14 | 14.0 | 0.93 | 14 | 14.0 | 0.745 |
| banknote | 4 | 4.0 | 1.6709 | 4 | 4.0 | 1.7985 |
| gender-by-voice | 20 | 20.0 | 0.0243 | 20 | 20.0 | 0.0296 |
| waveform | 40 | 40.0 | 0.8176 | 40 | 40.0 | 0.7742 |
| wind | 14 | 14.0 | 1.1833 | 14 | 14.0 | 0.8493 |
| speech | 400 | 400.0 | 1.6003 | 400 | 400.0 | 1.1995 |

Table 12: XGBoost Sign-OPT perturbations statistics

| | OPT $L_2$ | | | OPT $L_\infty$ | | |
|---|---|---|---|---|---|---|
| **Dataset name** | **Max #feature changed** | **Mean #feature changed** | **Mean $\|L_2\|$** | **Max #feature changed** | **Mean #feature changed** | **Mean $\|L_\infty\|$** |
| breast-cancer | 10 | 10.0 | 0.2288 | 10 | 10.0 | 0.2274 |
| covtype | 54 | 54.0 | 0.0568 | 54 | 54.0 | 0.0494 |
| cod-rna | 8 | 8.0 | 0.0436 | 8 | 8.0 | 0.039 |
| diabetes | 8 | 8.0 | 0.0462 | 8 | 8.0 | 0.0613 |
| Fashion-MNIST | 784 | 784.0 | 0.0973 | 784 | 784.0 | 0.1023 |
| ijcnn1 | 22 | 22.0 | 0.0519 | 22 | 22.0 | 0.0418 |
| MNIST | 784 | 784.0 | 0.089 | 784 | 784.0 | 0.1551 |
| sensorless | 48 | 48.0 | 0.0201 | 48 | 48.0 | 0.0233 |
| webspam | 254 | 254.0 | 0.0132 | 254 | 254.0 | 0.0095 |
| MNIST 2 vs. 6 | 784 | 784.0 | 0.2615 | 784 | 784.0 | 0.4055 |
| electricity | 8 | 8.0 | 0.0081 | 8 | 8.0 | 0.0107 |
| drybean | 16 | 16.0 | 0.0069 | 16 | 16.0 | 0.0481 |
| adult | 14 | 14.0 | 0.8486 | 14 | 14.0 | 0.7598 |
| banknote | 4 | 4.0 | 1.8506 | 4 | 4.0 | 1.3701 |
| gender-by-voice | 20 | 20.0 | 0.0288 | 20 | 20.0 | 0.0294 |
| waveform | 40 | 40.0 | 0.8439 | 40 | 40.0 | 0.7449 |
| wind | 14 | 14.0 | 0.886 | 14 | 14.0 | 0.876 |
| speech | 400 | 400.0 | 1.3453 | 400 | 400.0 | 1.6119 |

Table 13: XGBoost OPT perturbations statistics

| | HSJA $L_2$ | | | HSJA $L_\infty$ | | |
|---|---|---|---|---|---|---|
| **Dataset name** | **Max #feature changed** | **Mean #feature changed** | **Mean $\|L_2\|$** | **Max #feature changed** | **Mean #feature changed** | **Mean $\|L_\infty\|$** |
| breast-cancer | 10 | 9.8537 | 0.3335 | 10 | 9.8049 | 0.2454 |
| covtype | 53 | 43.42 | 0.1805 | 52 | 43.49 | 0.199 |
| cod-rna | 8 | 7.98 | 0.0854 | 8 | 8.0 | 0.1099 |
| diabetes | 8 | 7.9091 | 0.0546 | 8 | 7.8235 | 0.0668 |
| Fashion-MNIST | 773 | 736.66 | 1.8948 | 776 | 740.18 | 3.2664 |
| ijcnn1 | 22 | 20.09 | 0.0961 | 22 | 19.95 | 0.1031 |
| MNIST | 768 | 707.41 | 4.0607 | 762 | 705.16 | 1.0578 |
| sensorless | 48 | 48.0 | 0.0669 | 48 | 48.0 | 0.0747 |
| webspam | 253 | 231.35 | 0.1992 | 253 | 233.32 | 0.4914 |
| MNIST 2 vs. 6 | 769 | 709.69 | 14.8869 | 763 | 723.38 | 38.2944 |
| electricity | 8 | 7.78 | 0.0199 | 8 | 7.88 | 0.0185 |
| drybean | 16 | 14.46 | 1683.5416 | 16 | 14.45 | 2061.8314 |
| adult | 14 | 13.31 | 95.9496 | 14 | 13.42 | 383.9427 |
| banknote | 4 | 4.0 | 4.3192 | 4 | 4.0 | 4.3699 |
| gender-by-voice | 20 | 19.85 | 0.0441 | 20 | 19.79 | 0.046 |
| waveform | 40 | 40.0 | 4.5889 | 40 | 39.96 | 4.7256 |
| wind | 14 | 13.8 | 7.9845 | 14 | 13.77 | 9.142 |
| speech | 400 | 400.0 | 66.9408 | 400 | 400.0 | 86.5313 |

Table 14: XGBoost HSJA perturbations statistics

| | Cube $L_2$ | | | Cube $L_\infty$ | | |
|---|---|---|---|---|---|---|
| **Dataset name** | **Max #feature changed** | **Mean #feature changed** | **Mean $\|L_2\|$** | **Max #feature changed** | **Mean #feature changed** | **Mean $\|L_\infty\|$** |
| breast-cancer | 4 | 2.7778 | 0.3549 | 10 | 9.1875 | 0.6094 |
| covtype | 6 | 2.52 | 0.0556 | 39 | 31.66 | 0.0592 |
| cod-rna | 6 | 2.92 | 0.0657 | 8 | 7.77 | 0.1299 |
| diabetes | 4 | 2.2 | 0.0398 | 8 | 7.0 | 0.0687 |
| Fashion-MNIST | 79 | 16.05 | 0.0172 | 658 | 555.6 | 0.0154 |
| ijcnn1 | 10 | 4.02 | 0.071 | 21 | 16.3 | 0.0389 |
| MNIST | 31 | 11.2344 | 0.0087 | 485 | 445.4375 | 0.0055 |
| sensorless | 26 | 3.9259 | 0.0065 | 48 | 45.77 | 0.0049 |
| webspam | 18 | 9.44 | 0.0032 | 193 | 165.21 | 0.0038 |
| MNIST 2 vs. 6 | 55 | 25.69 | 0.1066 | 537 | 478.7344 | 0.1532 |
| electricity | 8 | 3.26 | 1.9474 | 8 | 7.6 | 3.1336 |
| drybean | 12 | 4.3333 | 35279.2292 | 16 | 12.93 | 65949.48 |
| adult | 11 | 6.1042 | 168527.4792 | 14 | 11.9667 | 178961.0333 |
| banknote | 3 | 1.6 | 3.2732 | 4 | 3.9474 | 5.7962 |
| gender-by-voice | 14 | 3.8788 | 2.9815 | 20 | 19.7368 | 46.5609 |
| waveform | 25 | 7.94 | 3.8412 | 40 | 39.3085 | 4.509 |
| wind | 14 | 5.7111 | 39.8756 | 14 | 13.9167 | 68.9444 |
| speech | - | - | - | - | - | - |

Table 15: XGBoost Cube perturbations statistics

| | Leaf-Tuple $L_2$ | | | Leaf-Tuple $L_\infty$ | | |
|---|---|---|---|---|---|---|
| **Dataset name** | **Max #feature changed** | **Mean #feature changed** | **Mean $\|L_2\|$** | **Max #feature changed** | **Mean #feature changed** | **Mean $\|L_\infty\|$** |
| breast-cancer | 8 | 5.85 | 0.2015 | 10 | 6.1282 | 0.228 |
| covtype | 7 | 2.717 | 0.044 | 9 | 3.7392 | 0.0404 |
| cod-rna | 8 | 7.9905 | 0.0399 | 8 | 7.9832 | 0.035 |
| diabetes | 7 | 4.25 | 0.048 | 7 | 4.6296 | 0.0556 |
| Fashion-MNIST | 703 | 420.8461 | 0.8065 | 715 | 429.9784 | 0.8169 |
| ijcnn1 | 12 | 11.999 | 0.0385 | 12 | 12.0 | 0.0338 |
| MNIST | 357 | 179.9864 | 0.8689 | 301 | 181.6677 | 0.8722 |
| sensorless | 16 | 4.2944 | 0.0244 | 20 | 6.2854 | 0.0209 |
| webspam | 34 | 17.6316 | 0.4336 | 31 | 18.2457 | 0.4364 |
| MNIST 2 vs. 6 | 324 | 196.8853 | 0.8665 | 297 | 193.2404 | 0.8729 |
| electricity | 7 | 2.8221 | 2.9138 | 7 | 3.5814 | 3.0857 |
| drybean | 16 | 16.0 | 39343.4913 | 16 | 15.9892 | 36040.6187 |
| adult | 13 | 11.179 | 188393.0498 | 12 | 10.9213 | 198588.9907 |
| banknote | 4 | 2.4746 | 4.2745 | 4 | 2.9167 | 4.4112 |
| gender-by-voice | 20 | 18.1538 | 55.5696 | 20 | 18.3013 | 31.3279 |
| waveform | 24 | 19.3467 | 3.9899 | 29 | 22.764 | 4.0157 |
| wind | 14 | 13.7389 | 67.9944 | 14 | 13.8936 | 68.1223 |
| speech | 99 | 69.6374 | 2.3355 | 100 | 71.9725 | 2.37 |

Table 16: XGBoost Leaf-Tuple perturbations statistics

## D.2 RANDOMFOREST PERTURBATION STATISTICS

| | Sign-OPT $L_2$ | | | Sign-OPT $L_\infty$ | | |
|---|---|---|---|---|---|---|
| Dataset name | Max #feature changed | Mean #feature changed | Mean $\|L_2\|$ perturbation | Max #feature changed | Mean #feature changed | Mean $\|L_\infty\|$ perturbation |
| breast-cancer | 10 | 10.0 | 0.3264 | 10 | 10.0 | 0.2304 |
| covtype | 54 | 54.0 | 0.0682 | 54 | 54.0 | 0.0699 |
| cod-rna | 8 | 8.0 | 0.0537 | 8 | 8.0 | 0.0514 |
| diabetes | 8 | 8.0 | 0.0845 | 8 | 8.0 | 0.0839 |
| Fashion-MNIST | 784 | 784.0 | 0.0358 | 784 | 784.0 | 0.0258 |
| ijcnn1 | 22 | 22.0 | 0.0776 | 22 | 22.0 | 0.059 |
| MNIST | 784 | 784.0 | 0.0161 | 784 | 784.0 | 0.0105 |
| sensorless | 48 | 48.0 | 0.018 | 48 | 48.0 | 0.0236 |
| webspam | 254 | 254.0 | 0.0032 | 254 | 254.0 | 0.0019 |
| MNIST 2 vs. 6 | 784 | 784.0 | 0.2535 | 784 | 784.0 | 0.076 |
| electricity | 8 | 8.0 | 0.0189 | 8 | 8.0 | 0.0177 |
| drybean | 16 | 16.0 | 0.0145 | 16 | 16.0 | 0.0119 |
| adult | 14 | 14.0 | 0.96 | 14 | 14.0 | 1.115 |
| banknote | 4 | 4.0 | 1.5134 | 4 | 4.0 | 1.4708 |
| gender-by-voice | 20 | 20.0 | 0.0294 | 20 | 20.0 | 0.027 |
| waveform | 40 | 40.0 | 0.9492 | 40 | 40.0 | 0.7961 |
| wind | 14 | 14.0 | 1.1805 | 14 | 14.0 | 1.1561 |
| speech | - | - | - | - | - | - |

Table 17: RandomForest Sign-OPT perturbations statistics

| | OPT $L_2$ | | | OPT $L_\infty$ | | |
|---|---|---|---|---|---|---|
| Dataset name | Max #feature changed | Mean #feature changed | Mean $\|L_2\|$ | Max #feature changed | Mean #feature changed | Mean $\|L_\infty\|$ |
| breast-cancer | 10 | 10.0 | 0.3403 | 10 | 10.0 | 0.2835 |
| covtype | 54 | 54.0 | 0.0691 | 54 | 54.0 | 0.0675 |
| cod-rna | 8 | 8.0 | 0.0604 | 8 | 8.0 | 0.0599 |
| diabetes | 8 | 8.0 | 0.0976 | 8 | 8.0 | 0.0989 |
| Fashion-MNIST | 784 | 784.0 | 0.0459 | 784 | 784.0 | 0.0444 |
| ijcnn1 | 22 | 22.0 | 0.0665 | 22 | 22.0 | 0.0641 |
| MNIST | 784 | 784.0 | 0.0253 | 784 | 784.0 | 0.0126 |
| sensorless | 48 | 48.0 | 0.0403 | 48 | 48.0 | 0.0266 |
| webspam | 254 | 254.0 | 0.0028 | 254 | 254.0 | 0.0027 |
| MNIST 2 vs. 6 | 784 | 784.0 | 0.1705 | 784 | 784.0 | 0.1734 |
| electricity | 8 | 8.0 | 0.0214 | 8 | 8.0 | 0.0185 |
| drybean | 16 | 16.0 | 0.0239 | 16 | 16.0 | 0.0247 |
| adult | 14 | 14.0 | 0.9111 | 14 | 14.0 | 0.785 |
| banknote | 4 | 4.0 | 1.5897 | 4 | 4.0 | 1.5036 |
| gender-by-voice | 20 | 20.0 | 0.0308 | 20 | 20.0 | 0.0271 |
| waveform | 40 | 40.0 | 1.0321 | 40 | 40.0 | 0.8934 |
| wind | 14 | 14.0 | 1.3465 | 14 | 14.0 | 0.9985 |
| speech | - | - | - | - | - | - |

Table 18: RandomForest OPT perturbations statistics

| | HSJA $L_2$ | | | HSJA $L_\infty$ | | |
|---|---|---|---|---|---|---|
| Dataset name | Max #feature changed | Mean #feature changed | Mean $\|L_2\|$ | Max #feature changed | Mean #feature changed | Mean $\|L_\infty\|$ |
| breast-cancer | 10 | 9.3714 | 0.4717 | 10 | 9.5926 | 0.6591 |
| covtype | 50 | 43.42 | 0.2901 | 54 | 43.36 | 0.2679 |
| cod-rna | - | - | - | - | - | - |
| diabetes | - | - | - | - | - | - |
| Fashion-MNIST | 783 | 724.22 | 3.6077 | 766 | 720.79 | 0.9142 |
| ijcnn1 | 22 | 19.98 | 0.1871 | 22 | 20.0 | 0.2578 |
| MNIST | 756 | 672.46 | 0.5509 | 753 | 669.0 | 0.5491 |
| sensorless | 48 | 48.0 | 0.089 | 48 | 48.0 | 0.0641 |
| webspam | 246 | 219.8 | 0.1553 | 254 | 218.81 | 0.1238 |
| MNIST 2 vs. 6 | 768 | 667.37 | 9.4532 | 763 | 692.16 | 1.8106 |
| electricity | 8 | 7.79 | 0.039 | 8 | 7.83 | 0.031 |
| drybean | 16 | 14.86 | 3.2014 | 16 | 14.55 | 4.4453 |
| adult | 14 | 13.15 | 9.2404 | 14 | 13.19 | 6.6426 |
| banknote | 4 | 4.0 | 3.7265 | 4 | 4.0 | 3.035 |
| gender-by-voice | 20 | 19.78 | 0.0537 | 20 | 19.79 | 0.0457 |
| waveform | 40 | 40.0 | 4.6663 | 40 | 39.98 | 3.8479 |
| wind | 14 | 13.67 | 10.386 | 14 | 13.76 | 7.8599 |
| speech | - | - | - | - | - | - |

Table 19: RandomForest HSJA perturbations statistics

| | Cube $L_2$ | | | Cube $L_\infty$ | | |
|---|---|---|---|---|---|---|
| **Dataset name** | **Max #feature changed** | **Mean #feature changed** | **Mean $||L_2||$** | **Max #feature changed** | **Mean #feature changed** | **Mean $||L_\infty||$** |
| breast-cancer | 7 | 2.5556 | 0.3326 | 10 | 9.2857 | 0.5759 |
| covtype | 6 | 1.44 | 0.1144 | 42 | 32.34 | 0.1195 |
| cod-rna | 6 | 2.03 | 0.0979 | 8 | 7.77 | 0.1724 |
| diabetes | 3 | 1.75 | 0.1375 | 8 | 7.2381 | 0.124 |
| Fashion-MNIST | 22 | 8.66 | 0.0348 | 715 | 552.09 | 0.0251 |
| ijcnn1 | 7 | 2.47 | 0.094 | 21 | 15.83 | 0.0804 |
| MNIST | 12 | 3.91 | 0.0122 | 563 | 461.19 | 0.0131 |
| sensorless | 9 | 2.56 | 0.0306 | 48 | 44.74 | 0.0235 |
| webspam | 7 | 2.83 | 0.0043 | 203 | 164.43 | 0.0042 |
| MNIST 2 vs. 6 | 16 | 7.03 | 0.1644 | 581 | 472.78 | 0.1066 |
| electricity | 5 | 1.6 | 0.0972 | 8 | 7.74 | 3.3875 |
| drybean | 9 | 2.68 | 3976.0933 | 16 | 15.66 | 53221.8 |
| adult | 2 | 1.27 | 9019.91 | 12 | 11.24 | 186698.07 |
| banknote | 3 | 2.0 | 5.9102 | 4 | 4.0 | 4.8118 |
| gender-by-voice | 15 | 4.8636 | 2.4741 | 20 | 19.6341 | 27.7117 |
| waveform | 30 | 9.2778 | 3.8403 | 40 | 34.6364 | 4.5004 |
| wind | - | - | - | - | - | - |
| speech | - | - | - | - | - | - |

Table 20: RandomForest Cube perturbations statistics

| | Leaf-Tuple $L_2$ | | | Leaf-Tuple $L_\infty$ | | |
|---|---|---|---|---|---|---|
| **Dataset name** | **Max #feature changed** | **Mean #feature changed** | **Mean $||L_2||$** | **Max #feature changed** | **Mean #feature changed** | **Mean $||L_\infty||$** |
| breast-cancer | 10 | 6.3902 | 0.2507 | 9 | 5.8095 | 0.2807 |
| covtype | - | - | - | - | - | - |
| cod-rna | 8 | 7.9845 | 0.0558 | 8 | 7.9922 | 0.0557 |
| diabetes | - | - | - | - | - | - |
| Fashion-MNIST | 655 | 424.3636 | 0.8172 | 666 | 379.701 | 0.0023 |
| ijcnn1 | 12 | 12.0 | 0.051 | 12 | 12.0 | 0.0568 |
| MNIST | 276 | 184.2766 | 0.8699 | 303 | 184.1649 | 0.8457 |
| sensorless | 2 | 1.0385 | 0.0002 | 3 | 1.1168 | 0.0001 |
| webspam | 23 | 11.7593 | 0.4215 | 28 | 13.0909 | 0.4181 |
| MNIST 2 vs. 6 | 324 | 190.5356 | 0.8724 | 324 | 191.279 | 0.8709 |
| electricity | 6 | 2.7642 | 3.0778 | 7 | 3.5 | 3.1415 |
| drybean | 16 | 16.0 | 51659.5417 | 16 | 16.0 | 50043.6164 |
| adult | 12 | 10.5597 | 199668.9416 | 12 | 10.7183 | 192389.8652 |
| banknote | 4 | 2.8361 | 5.1703 | 4 | 3.1034 | 5.3954 |
| gender-by-voice | 19 | 18.1154 | 34.3206 | 20 | 18.1325 | 32.0345 |
| waveform | 23 | 17.78 | 4.1968 | 25 | 18.5915 | 4.4844 |
| wind | 14 | 13.7556 | 67.8056 | 14 | 13.733 | 68.983 |
| speech | - | - | - | - | - | - |

Table 21: RandomForest Leaf-Tuple perturbations statistics

# E    APPENDIX E - DETECTOR ABLATION STUDY

We wanted to investigate how three aspects of our detector affect our metrics results:

1.  The dimensions of our new representations (for the samples and for the nodes).

2.  The machine learning classifier that was chosen as the final adversarial detector.

3.  Key hyperparameters of the final adversarial detector.

To do our tests, we took two different experiments and showed their results:

- XGBoost target with codrna dataset and OPT attack method.

- RandomForest target with sensorless dataset and HopSkipJumpAttack attack method.

Each one of the above with the two different norms.

## E.1    DIFFERENT CLASSIFIER & HYPERPARAMETERS

We checked three different classifiers as our models for our adversarial evasion attack detector: XGBoost, RandomForest, and K-nearest neighbors (KNN). For the tree ensemble classifiers, we checked how much the number of estimators and their depth impacts the performance of the detector. For the KNN classifier, we tested the results with different values of K.

### E.1.1    XGBOOST DETECTOR HYPERPARAMETERS

As we can see from the results in Figures 13, 14, 15, and 16 that in general the number of estimators has a large impact on both ROC-AUC and on PR-AUC until a certain point which is around 50 estimators and above usually there is improvements, but relatively smaller. In most cases, the maximum depth of the trees has a very light impact on the results when the number of estimators is 50 or more.

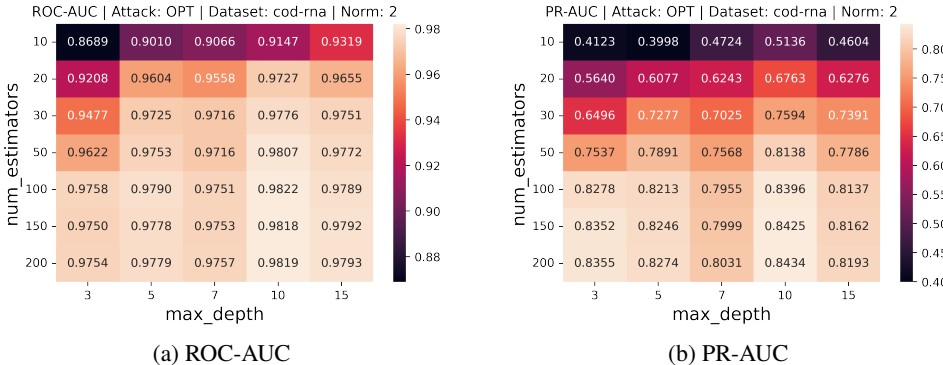

(a) ROC-AUC                                    (b) PR-AUC

Figure 13: Comparing ROC-AUC and PR-AUC for detector based on XGBoost classifier with different hyperparametrs. Target: XGBoost, dataset: codrna, attack method: OPT, norm: 2.

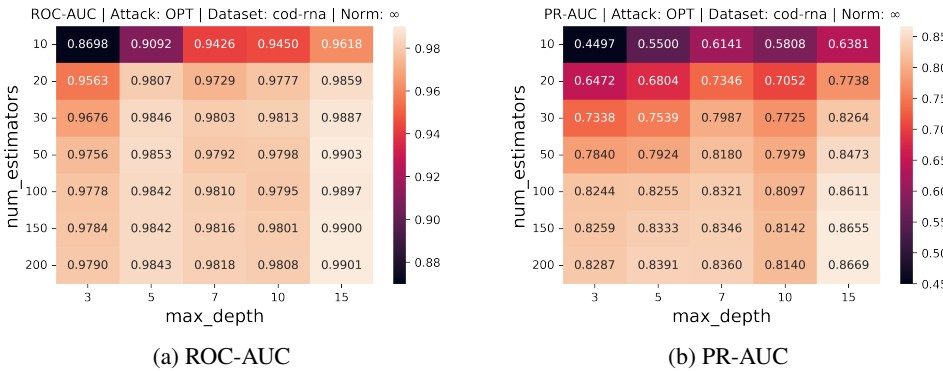

(a) ROC-AUC                    (b) PR-AUC

Figure 14: Comparing ROC-AUC and PR-AUC for detector based on XGBoost classifier with different hyperparametrs. Target: XGBoost, dataset: codrna, attack method: OPT, norm: $\infty$

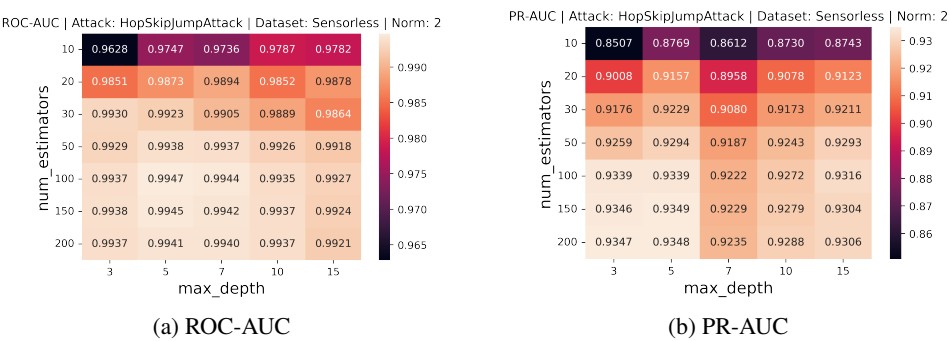

(a) ROC-AUC                    (b) PR-AUC

Figure 15: Comparing ROC-AUC and PR-AUC for detector based on XGBoost classifier with different hyperparametrs. Target: RandomForest, dataset: sensorless, attack method: HopSkipJumpAttack, norm: 2

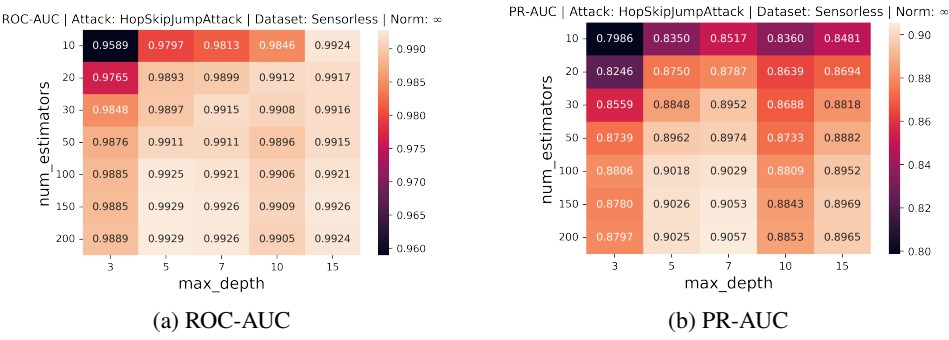

(a) ROC-AUC                    (b) PR-AUC

Figure 16: Comparing ROC-AUC and PR-AUC for detector based on XGBoost classifier with different hyperparameters. Target: RandomForest, dataset: sensorless, attack method: HopSkipJumpAttack, norm: $\infty$

### E.1.2 RANDOMFOREST

As we can see from the results in Figures 17, 18, 19, and 20 that in general the number of estimators and the maximum depth of the trees has a large impact on both ROC-AUC and on PR-AUC. In

experiments with hyperparameters similar to our XGBoost experiments both ROC-AUC and PR-AUC is lower.

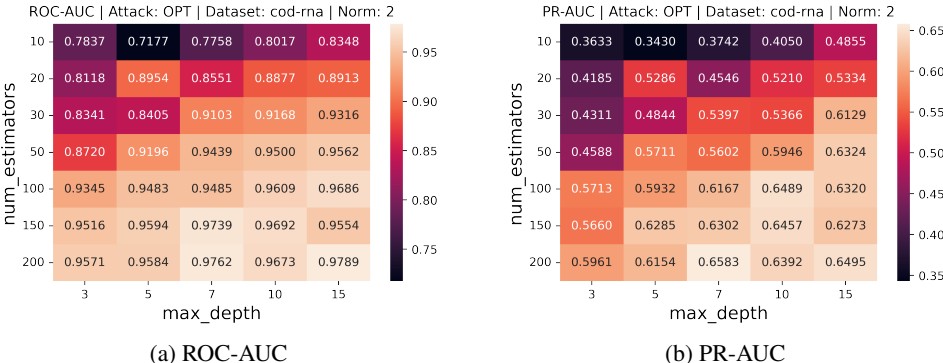

(a) ROC-AUC                    (b) PR-AUC

Figure 17: Comparing ROC-AUC and PR-AUC for detector based on RandomForest classifier with different hyperparametrs. Target: XGBoost, dataset: codrna, attack method: OPT, norm: 2

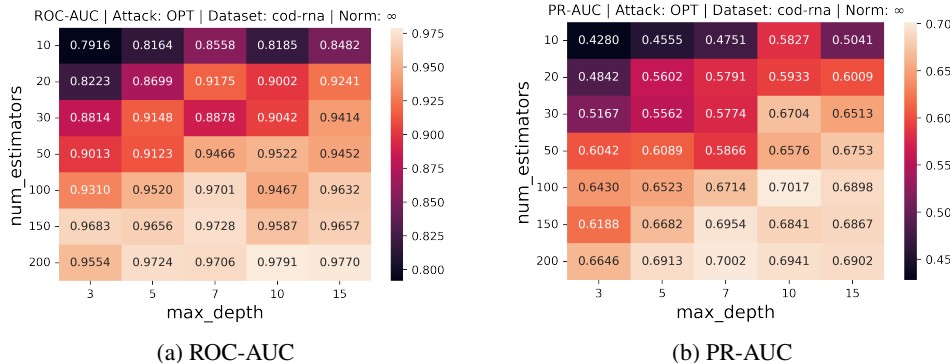

(a) ROC-AUC                    (b) PR-AUC

Figure 18: Comparing ROC-AUC and PR-AUC for detector based on RandomForest classifier with different hyperparameters. Target: XGBoost, dataset: codrna, attack method: OPT, norm: ∞

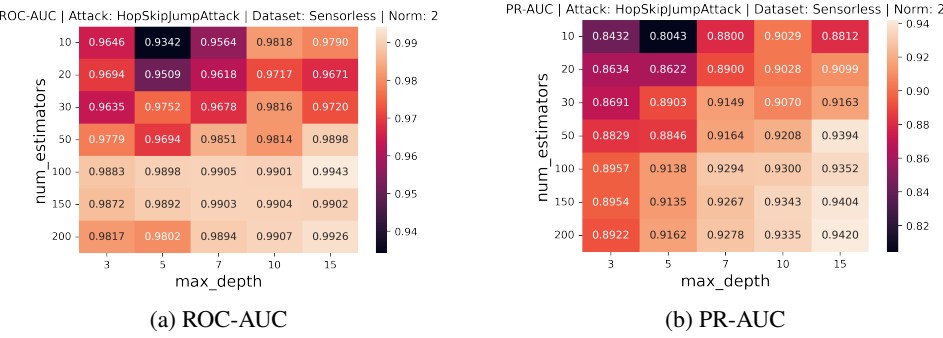

(a) ROC-AUC                    (b) PR-AUC

Figure 19: Comparing ROC-AUC and PR-AUC for detector based on RandomForest classifier with different hyperparameters. Target: RandomForest, dataset: sensorless, attack method: Hop-SkipJumpAttack, norm: 2

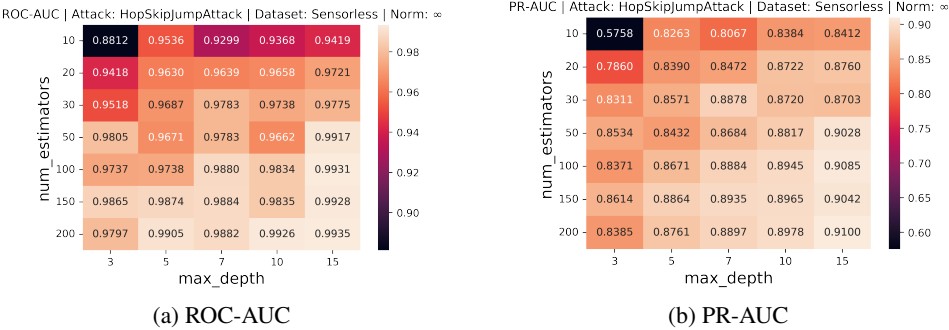

(a) ROC-AUC

(b) PR-AUC

Figure 20: Comparing ROC-AUC and PR-AUC for detector based on RandomForest classifier with different hyperparameters. Target: RandomForest, dataset: sensorless, attack method: Hop-SkipJumpAttack, norm: ∞

### E.1.3 KNN

We tested KNN performance with different Ks to investigate how much it changed the detector performance metrics and compared it to the results we got in our original experiments. We used KNN based on Facebook AI fast similarity search (Johnson et al., 2019) implemented in DESlib (Cruz et al., 2020) due to time consideration for cases where the feature count is relatively high. In Figures 21 and 22 we can see a comparison of ROC-AUC and PR-AUC of KNN classifier with different values of k neighbors. Additionally, added the performance of KNN on the original representation of the dataset and the metric values of the original experiments done with our new method with XGBoost classifier as the detector. In all of the experiments, we can see the relative stability of the values with different K values. Additionally, the metric values with the XGBoost classifier are higher.

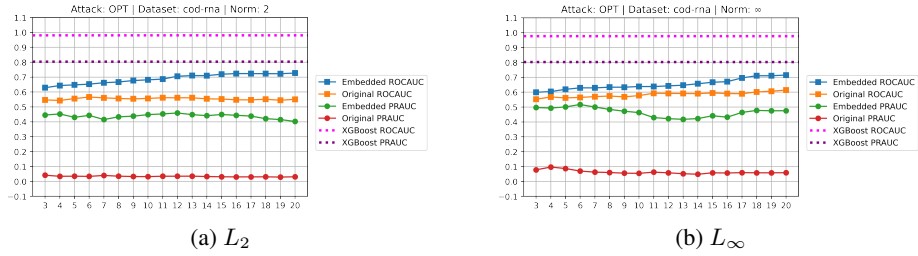

(a) $L_2$

(b) $L_\infty$

Figure 21: KNN classifier adversarial detector performance. Attack: OPT — Dataset: codrna.

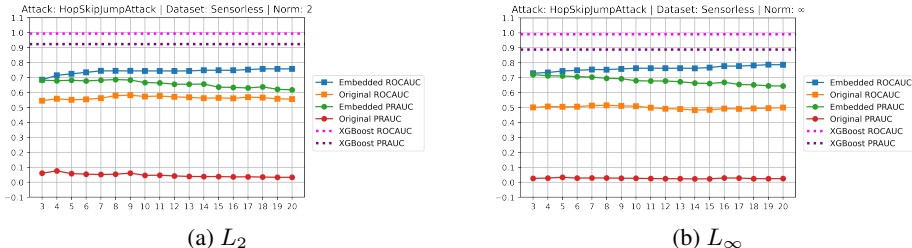

(a) $L_2$

(b) $L_\infty$

Figure 22: KNN classifier adversarial detector performance. Attack: HopSkipJumpAttack — Dataset: sensorless

## E.2 EMBEDDINGS DIMENSIONS

Here we tested how much the embedding dimension might affect the performance of our detector. We have two different embeddings - for the samples and for the nodes. In Figures 23, 24, 25, and 26 we can see heatmaps for compare the performance with different embedding size. The general pattern from the heatmaps in Figures 23 and 24 is that as our sample hidden size is larger and the node hidden size is lower the results are better. In Figures 25 and 26 we don't see that pattern.

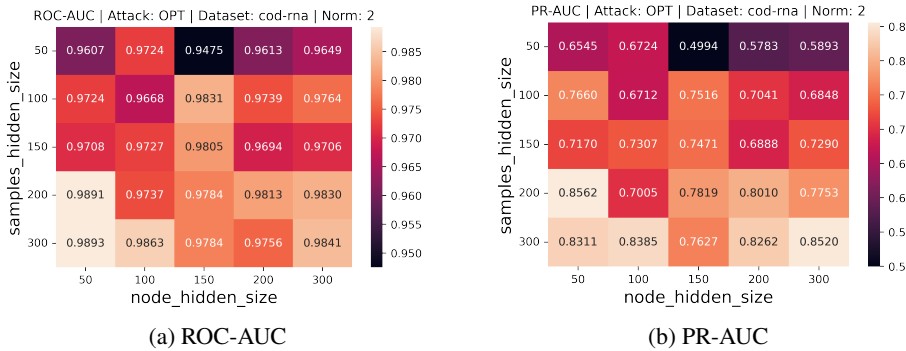

Figure 23: Performance of different samples and nodes embedding sizes. Target: XGBoost, dataset: codrna, attack method: OPT, norm: 2

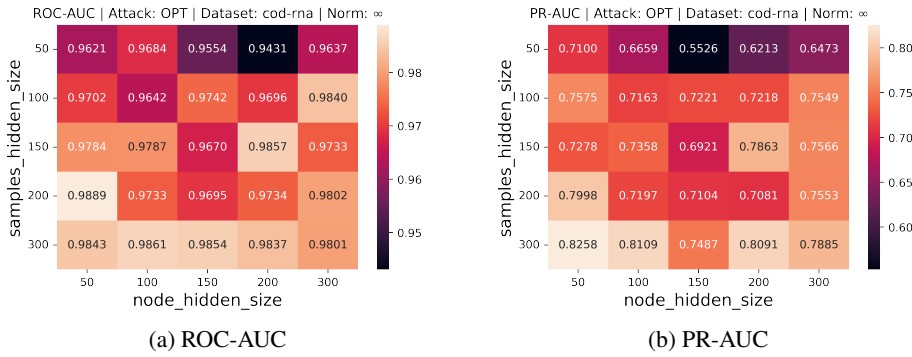

Figure 24: Performance of different samples and nodes embedding sizes. Target: XGBoost, dataset: codrna, attack method: OPT, norm: ∞

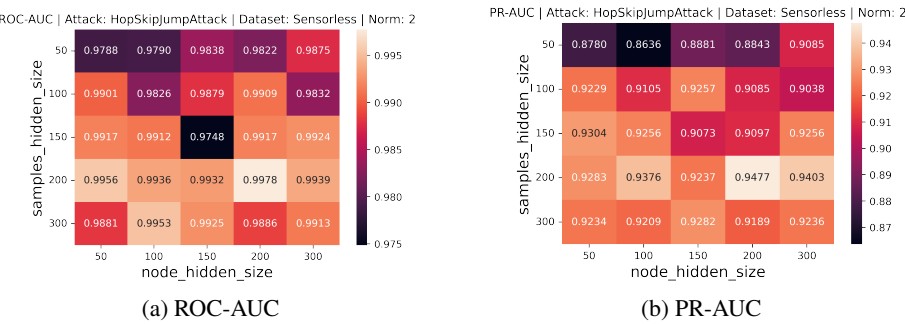

Figure 25: Performance of different samples and nodes embedding sizes. Target: RandomForest, dataset: sensorless, attack method: HSJA, norm: 2

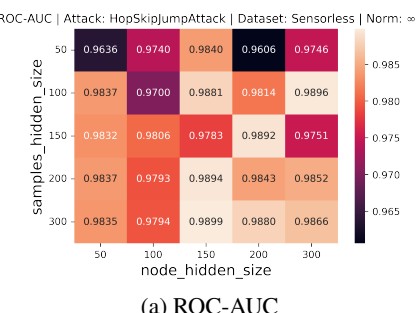
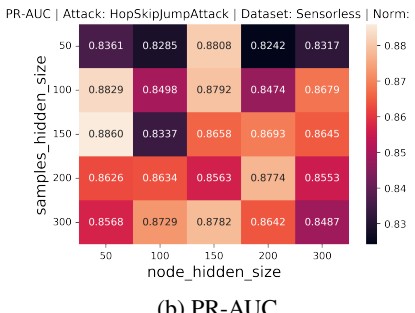

(a) ROC-AUC            (b) PR-AUC

Figure 26: Performance of different samples and nodes embedding sizes. Target: RandomForest, dataset: sensorless, attack method: HSJA, norm: $\infty$

## F  APPENDIX F - TREE ENSEMBLE HYPERPARAMETERS

We used tree ensembles for two tasks: the target model to extract adversarial samples and the base for the adversarial detector. For the target model, we used 40 estimators with a max depth of 5 for both XGBoost and Random forest. For the rest of the hyperparameters, we used the default values set in the XGBoost library package version 1.6.1. To train the adversarial detector, we used the default parameters for XGBoost and RandomForest set in the XGBoost library package version 1.6.1.

## G  APPENDIX G - TREE MODEL'S PERFORMANCE USING LESS DATA

A potential issue in our method is that we reduce the available data used to train the model; thus, the model performance might be affected. As part of our method, we split each dataset we experimented on into several subsets for different roles described in Subsection 4.1 and Appendix H. We took the original representation of the datasets and compared the trained models from our experiments to a new version of the model trained on all data besides the test set. Below, we can see in Figure 27 a comparison between the ROC-AUC of the model trained on our split version of the dataset (Y-axis) to the same model trained on more data (X-axis) on the test sets. As we can see, there is a decrease in performance most of the time, but most of the degradation is minor. For a complete description of the numbers in every experiment, refer to Tables 22, 23, 24, 25, 26, 27, 28, 29, 30, 31.

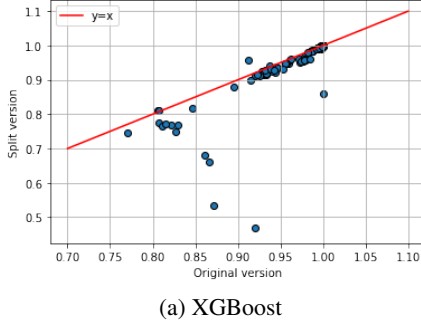
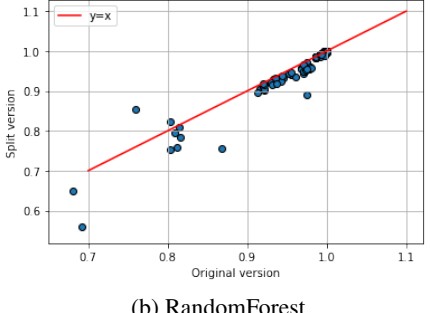

(a) XGBoost            (b) RandomForest

Figure 27: A comparison between our method models' ROC-AUC on the test set compared to a model trained with the same hyperparameters but with more data which we use in our method to train the representations and detector.

| Dataset | Sign-OPT $L_2$ | | | Sign-OPT $L_\infty$ | | |
|---|---|---|---|---|---|---|
| | Splitted Dataset | Original Dataset | $\Delta$ROC-AUC | Splitted Dataset | Original Dataset | $\Delta$ROC-AUC |
| breast_cancer | 0.9958 | 0.9975 | 0.0017 | 1.0 | 0.9989 | -0.0011 |
| covtype | 0.9843 | 0.9865 | 0.0022 | 0.9847 | 0.9868 | 0.0021 |
| codrna | 0.9928 | 0.994 | 0.0012 | 0.9928 | 0.994 | 0.0012 |
| diabetes | 0.7706 | 0.8154 | 0.0448 | 0.8108 | 0.8075 | -0.0033 |
| fashion | 0.9903 | 0.9924 | 0.0021 | 0.9899 | 0.9925 | 0.0026 |
| ijcnn1 | 0.9954 | 0.9961 | 0.0007 | 0.9951 | 0.9962 | 0.0011 |
| mnist | 0.9991 | 0.9997 | 0.0006 | 0.9991 | 0.9997 | 0.0006 |
| mnist26 | 0.9995 | 0.9999 | 0.0004 | 0.9997 | 0.9999 | 0.0002 |
| sensorless | 1.0 | 1.0 | 0.0 | 1.0 | 1.0 | 0.0 |
| webspam | 0.9989 | 0.9992 | 0.0003 | 0.9989 | 0.9992 | 0.0003 |
| electricity | 0.9614 | 0.9783 | 0.0169 | 0.9579 | 0.9782 | 0.0203 |
| drybean | 0.9957 | 0.9964 | 0.0007 | 0.9941 | 0.9953 | 0.0012 |
| adult | 0.913 | 0.9254 | 0.0124 | 0.9149 | 0.9235 | 0.0086 |
| banknote | 1.0 | 0.9992 | -0.0008 | 0.9983 | 1.0 | 0.0017 |
| voice | 0.9957 | 0.9992 | 0.0035 | 0.9806 | 0.9817 | 0.0011 |
| waveform | 0.962 | 0.9847 | 0.0227 | 0.9274 | 0.9322 | 0.0048 |
| wind | 0.9301 | 0.9391 | 0.009 | 0.9294 | 0.9424 | 0.013 |
| speech | 0.959 | 0.9126 | -0.0464 | 0.4699 | 0.9208 | 0.4509 |

Table 22: Sign-OPT XGBoost Experiments - ROC-AUC Degradation

| Dataset | OPT $L_2$ | | | OPT $L_\infty$ | | |
|---|---|---|---|---|---|---|
| | Splitted Dataset | Original Dataset | $\Delta$ROC-AUC | Splitted Dataset | Original Dataset | $\Delta$ROC-AUC |
| breast_cancer | 0.9958 | 0.9969 | 0.0011 | 0.9978 | 0.9978 | 0.0 |
| covtype | 0.9846 | 0.9868 | 0.0022 | 0.9845 | 0.9864 | 0.0019 |
| codrna | 0.9926 | 0.9939 | 0.0013 | 0.9929 | 0.9941 | 0.0012 |
| diabetes | 0.7669 | 0.8217 | 0.0548 | 0.8103 | 0.8055 | -0.0048 |
| fashion | 0.99 | 0.9924 | 0.0024 | 0.9903 | 0.9926 | 0.0023 |
| ijcnn1 | 0.9942 | 0.996 | 0.0018 | 0.9941 | 0.9959 | 0.0018 |
| mnist | 0.9992 | 0.9997 | 0.0005 | 0.9992 | 0.9997 | 0.0005 |
| mnist26 | 0.9997 | 0.9999 | 0.0002 | 0.9997 | 0.9999 | 0.0002 |
| sensorless | 1.0 | 1.0 | 0.0 | 1.0 | 1.0 | 0.0 |
| webspam | 0.9989 | 0.9992 | 0.0003 | 0.999 | 0.9993 | 0.0003 |
| electricity | 0.9529 | 0.9767 | 0.0238 | 0.9546 | 0.9735 | 0.0189 |
| drybean | 0.9945 | 0.9967 | 0.0022 | 0.9951 | 0.9967 | 0.0016 |
| adult | 0.9191 | 0.9291 | 0.01 | 0.8808 | 0.8952 | 0.0144 |
| banknote | 1.0 | 1.0 | 0.0 | 1.0 | 1.0 | 0.0 |
| voice | 0.9986 | 0.9997 | 0.0011 | 1.0 | 1.0 | 0.0 |
| waveform | 0.9287 | 0.9341 | 0.0054 | 0.9662 | 0.9777 | 0.0115 |
| wind | 0.9258 | 0.9395 | 0.0137 | 0.9171 | 0.9336 | 0.0165 |
| speech | 0.6603 | 0.8665 | 0.2062 | 0.8167 | 0.8467 | 0.03 |

Table 23: OPT XGBoost Experiments - ROC-AUC Degradation

| Dataset | HSJA $L_2$ | | | HSJA $L_\infty$ | | |
|---|---|---|---|---|---|---|
| | Splitted Dataset | Original Dataset | $\Delta$ROC-AUC | Splitted Dataset | Original Dataset | $\Delta$ROC-AUC |
| breast_cancer | 0.9994 | 0.9969 | -0.0025 | 0.9997 | 0.9989 | -0.0005 |
| covtype | 0.9851 | 0.9856 | 0.0005 | 0.9843 | 0.9867 | 0.0024 |
| codrna | 0.9929 | 0.9941 | 0.0012 | 0.9927 | 0.9939 | 0.0012 |
| diabetes | 0.7996 | 0.8215 | 0.0219 | 0.8119 | 0.82 | 0.0204 |
| fashion | 0.99 | 0.9925 | 0.0025 | 0.9901 | 0.9926 | 0.0025 |
| ijcnn1 | 0.9949 | 0.9956 | 0.0007 | 0.9945 | 0.9959 | 0.0014 |
| mnist | 0.9991 | 0.9997 | 0.0006 | 0.9992 | 0.9997 | 0.0005 |
| mnist26 | 0.9996 | 0.9999 | 0.0003 | 0.9998 | 0.9999 | 0.0001 |
| sensorless | 1.0 | 1.0 | 0.0 | 1.0 | 1.0 | 0.0 |
| webspam | 0.9989 | 0.9992 | 0.0003 | 0.9989 | 0.9992 | 0.0003 |
| electricity | 0.9526 | 0.9733 | 0.0207 | 0.9567 | 0.98 | 0.0233 |
| $dry_bean$ | 0.9924 | 0.9949 | 0.0025 | 0.9939 | 0.9956 | 0.0017 |
| adult | 0.9103 | 0.9196 | 0.0093 | 0.915 | 0.9328 | 0.0178 |
| banknote | 0.9966 | 1.0 | 0.0034 | 1.0 | 1.0 | 0.0 |
| voice | 0.994 | 0.996 | 0.002 | 0.9986 | 0.9995 | 0.0009 |
| waveform | 0.9605 | 0.9627 | 0.0022 | 0.9466 | 0.9574 | 0.0108 |
| wind | 0.9246 | 0.9282 | 0.0036 | 0.9347 | 0.9463 | 0.0116 |
| speech | 0.7473 | 0.7711 | 0.0238 | 0.8587 | 1.0 | 0.1413 |

Table 24: HSJA XGBoost Experiments - ROC-AUC Degradation

| | Cube $L_2$ | | | Cube $L_\infty$ | | |
|---|---|---|---|---|---|---|
| Dataset | Splitted Dataset | Original Dataset | ΔROC-AUC | Splitted Dataset | Original Dataset | ΔROC-AUC |
| breast_cancer | 0.9997 | 0.998 | -0.0017 | 0.9964 | 0.9966 | 0.0002 |
| covtype | 0.9852 | 0.9873 | 0.0021 | 0.9847 | 0.9862 | 0.0015 |
| codrna | 0.9928 | 0.994 | 0.0012 | 0.9929 | 0.9941 | 0.0012 |
| diabetes | 0.7477 | 0.8268 | 0.0791 | 0.7765 | 0.8072 | 0.0307 |
| fashion | 0.9903 | 0.9925 | 0.0022 | 0.99 | 0.9927 | 0.0027 |
| ijcnn1 | 0.9948 | 0.9959 | 0.0011 | 0.9945 | 0.9959 | 0.0014 |
| mnist | 0.9992 | 0.9997 | 0.0005 | 0.9992 | 0.9997 | 0.0005 |
| mnist26 | 0.9997 | 1.0 | 0.0003 | 0.9995 | 0.9999 | 0.0004 |
| sensorless | 1.0 | 1.0 | 0.0 | 1.0 | 1.0 | 0.0 |
| webspam | 0.9989 | 0.9993 | 0.0004 | 0.9989 | 0.9993 | 0.0004 |
| electricity | 0.9619 | 0.9801 | 0.0182 | 0.9528 | 0.9746 | 0.0218 |
| drybean | 0.9935 | 0.9951 | 0.0016 | 0.9942 | 0.9956 | 0.0014 |
| adult | 0.9153 | 0.9263 | 0.011 | 0.9142 | 0.9302 | 0.016 |
| banknote | 1.0 | 1.0 | 0.0 | 1.0 | 1.0 | 0.0 |
| voice | 1.0 | 1.0 | 0.0 | 0.9949 | 0.9973 | 0.0024 |
| waveform | 0.9475 | 0.9599 | 0.0124 | 0.9581 | 0.9612 | 0.0031 |
| wind | 0.9373 | 0.9451 | 0.0078 | 0.9425 | 0.937 | -0.0055 |
| speech | - | - | - | - | - | - |

Table 25: Cube XGBoost Experiments - ROC-AUC Degradation

| | Leaf-Tuple $L_2$ | | | Leaf-Tuple $L_\infty$ | | |
|---|---|---|---|---|---|---|
| Dataset | Splitted Dataset | Original Dataset | ΔROC-AUC | Splitted Dataset | Original Dataset | ΔROC-AUC |
| breast_cancer | 0.9986 | 0.9978 | -0.0008 | 0.9961 | 0.9972 | 0.0011 |
| covtype | 0.9845 | 0.9868 | 0.0023 | 0.9842 | 0.9869 | 0.0027 |
| codrna | 0.993 | 0.994 | 0.001 | 0.9928 | 0.9939 | 0.0011 |
| diabetes | 0.7662 | 0.8114 | 0.0452 | 0.7695 | 0.8298 | 0.0603 |
| fashion | 0.9902 | 0.9923 | 0.0021 | 0.99 | 0.9925 | 0.0025 |
| ijcnn1 | 0.9953 | 0.9963 | 0.001 | 0.9944 | 0.9963 | 0.0019 |
| mnist | 0.9992 | 0.9997 | 0.0005 | 0.9992 | 0.9997 | 0.0005 |
| mnist26 | 0.9997 | 0.9999 | 0.0002 | 0.9997 | 0.9999 | 0.0002 |
| sensorless | 1.0 | 1.0 | 0.0 | 1.0 | 1.0 | 0.0 |
| webspam | 0.9989 | 0.9993 | 0.0004 | 0.999 | 0.9993 | 0.0003 |
| electricity | 0.9553 | 0.9726 | 0.0173 | 0.9509 | 0.9732 | 0.0223 |
| drybean | 0.9956 | 0.9962 | 0.0006 | 0.9962 | 0.9963 | 0.0001 |
| adult | 0.9161 | 0.928 | 0.0119 | 0.9004 | 0.9147 | 0.0143 |
| banknote | 1.0 | 1.0 | 0.0 | 1.0 | 1.0 | 0.0 |
| voice | 0.9914 | 0.9971 | 0.0057 | 0.9919 | 0.9948 | 0.0029 |
| waveform | 0.9598 | 0.9716 | 0.0118 | 0.9487 | 0.9556 | 0.0069 |
| wind | 0.9328 | 0.9529 | 0.0201 | 0.9215 | 0.9434 | 0.0219 |
| speech | 0.5328 | 0.8716 | 0.3388 | 0.6808 | 0.8616 | 0.1808 |

Table 26: Leaf-Tuple XGBoost Experiments - ROC-AUC Degradation

| | Sign-OPT $L_2$ | | | Sign-OPT $L_\infty$ | | |
|---|---|---|---|---|---|---|
| Dataset | Splitted Dataset | Original Dataset | ΔROC-AUC | Splitted Dataset | Original Dataset | ΔROC-AUC |
| breast_cancer | 0.9952 | 0.9972 | 0.002 | 0.9965 | 0.9975 | 0.001 |
| covtype | 0.9841 | 0.9862 | 0.0021 | 0.9844 | 0.9864 | 0.002 |
| codrna | 0.9926 | 0.994 | 0.0014 | 0.9929 | 0.994 | 0.0011 |
| diabetes | 0.7954 | 0.8083 | 0.0129 | 0.7848 | 0.8162 | 0.0314 |
| fashion | 0.9904 | 0.9924 | 0.002 | 0.9901 | 0.9925 | 0.0024 |
| ijcnn1 | 0.9946 | 0.9964 | 0.0018 | 0.9956 | 0.9958 | 0.0002 |
| mnist | 0.9993 | 0.9997 | 0.0004 | 0.9992 | 0.9997 | 0.0005 |
| mnist26 | 0.9995 | 0.9999 | 0.0004 | 0.9997 | 0.9999 | 0.0002 |
| sensorless | 1.0 | 1.0 | 0.0 | 1.0 | 1.0 | 0.0 |
| webspam | 0.9989 | 0.9993 | 0.0004 | 0.9989 | 0.9993 | 0.0004 |
| electricity | 0.9525 | 0.9741 | 0.0216 | 0.9541 | 0.9776 | 0.0235 |
| drybean | 0.9968 | 0.9967 | -0.0001 | 0.9946 | 0.9952 | 0.0006 |
| adult | 0.9037 | 0.9209 | 0.0172 | 0.9146 | 0.9218 | 0.0072 |
| banknote | 1.0 | 1.0 | 0.0 | 1.0 | 1.0 | 0.0 |
| voice | 0.9894 | 0.9975 | 0.0081 | 0.9921 | 0.993 | 0.0009 |
| waveform | 0.957 | 0.98 | 0.023 | 0.9535 | 0.9748 | 0.0213 |
| wind | 0.9179 | 0.9196 | 0.0017 | 0.9176 | 0.9366 | 0.019 |
| speech | - | - | - | - | - | - |

Table 27: Sign-OPT RandomForest Experiments - ROC-AUC Degradation

| Dataset | OPT $L_2$ | | | OPT $L_\infty$ | | |
|---|---|---|---|---|---|---|
| | Splitted Dataset | Original Dataset | $\Delta$ROC-AUC | Splitted Dataset | Original Dataset | $\Delta$ROC-AUC |
| breast_cancer | 0.9992 | 0.9972 | -0.002 | 0.9994 | 0.9975 | -0.0019 |
| covtype | 0.9844 | 0.9868 | 0.0024 | 0.9851 | 0.9867 | 0.0016 |
| codrna | 0.9929 | 0.994 | 0.0011 | 0.9928 | 0.9941 | 0.0013 |
| diabetes | 0.7532 | 0.8033 | 0.0501 | 0.8241 | 0.8033 | -0.0208 |
| fashion | 0.9903 | 0.9924 | 0.0021 | 0.9902 | 0.9924 | 0.0022 |
| ijcnn1 | 0.9943 | 0.9958 | 0.0015 | 0.9958 | 0.996 | 0.0002 |
| mnist | 0.9993 | 0.9997 | 0.0004 | 0.9992 | 0.9997 | 0.0005 |
| mnist26 | 0.9998 | 0.9999 | 0.0001 | 0.9997 | 0.9999 | 0.0002 |
| sensorless | 1.0 | 1.0 | 0.0 | 1.0 | 1.0 | 0.0 |
| webspam | 0.999 | 0.9992 | 0.0002 | 0.9989 | 0.9992 | 0.0003 |
| electricity | 0.9595 | 0.9787 | 0.0192 | 0.9502 | 0.971 | 0.0208 |
| drybean | 0.9949 | 0.9958 | 0.0009 | 0.9962 | 0.9969 | 0.0007 |
| adult | 0.9147 | 0.9307 | 0.016 | 0.9106 | 0.9207 | 0.0101 |
| banknote | 1.0 | 1.0 | 0.0 | 1.0 | 1.0 | 0.0 |
| voice | 0.999 | 1.0 | 0.001 | 0.996 | 0.999 | 0.003 |
| waveform | 0.9652 | 0.9705 | 0.0053 | 0.9442 | 0.9519 | 0.0077 |
| wind | 0.9389 | 0.9436 | 0.0047 | 0.9452 | 0.9553 | 0.0101 |
| speech | - | - | - | - | - | - |

Table 28: OPT RandomForest Experiments - ROC-AUC Degradation

| Dataset | HSJA $L_2$ | | | HSJA $L_\infty$ | | |
|---|---|---|---|---|---|---|
| | Splitted Dataset | Original Dataset | $\Delta$ROC-AUC | Splitted Dataset | Original Dataset | $\Delta$ROC-AUC |
| breast_cancer | 0.998 | 0.9975 | -0.0005 | 0.9997 | 0.9983 | -0.0014 |
| covtype | 0.9843 | 0.9865 | 0.0022 | 0.9847 | 0.9865 | 0.0018 |
| codrna | - | - | - | - | - | - |
| diabetes | - | - | - | - | - | - |
| fashion | 0.9903 | 0.9925 | 0.0022 | 0.9901 | 0.9925 | 0.0024 |
| ijcnn1 | 0.9944 | 0.9963 | 0.0019 | 0.9951 | 0.9955 | 0.0004 |
| mnist | 0.9992 | 0.9997 | 0.0005 | 0.9992 | 0.9997 | 0.0005 |
| mnist26 | 0.9995 | 1.0 | 0.0005 | 0.9997 | 0.9999 | 0.0002 |
| sensorless | 1.0 | 1.0 | 0.0 | 1.0 | 1.0 | 0.0 |
| webspam | 0.9989 | 0.9992 | 0.0003 | 0.9989 | 0.9992 | 0.0003 |
| electricity | 0.9575 | 0.9736 | 0.0161 | 0.957 | 0.976 | 0.019 |
| dry$_b$ean | 0.995 | 0.995 | 0.0 | 0.9943 | 0.9964 | 0.0021 |
| adult | 0.9049 | 0.9154 | 0.0105 | 0.8973 | 0.9133 | 0.016 |
| banknote | 0.9966 | 1.0 | 0.0034 | 1.0 | 1.0 | 0.0 |
| voice | 1.0 | 1.0 | 0.0 | 0.999 | 1.0 | 0.001 |
| waveform | 0.9412 | 0.9548 | 0.0136 | 0.9329 | 0.945 | 0.0121 |
| wind | 0.9345 | 0.9432 | 0.0087 | 0.9292 | 0.9321 | 0.0029 |
| speech | - | - | - | - | - | - |

Table 29: HSJA RandomForest Experiments - ROC-AUC Degradation

| Dataset | Cube $L_2$ | | | Cube $L_\infty$ | | |
|---|---|---|---|---|---|---|
| | Splitted Dataset | Original Dataset | $\Delta$ROC-AUC | Splitted Dataset | Original Dataset | $\Delta$ROC-AUC |
| breast_cancer | 0.998 | 0.9969 | -0.0011 | 1.0 | 0.9964 | -0.0036 |
| covtype | 0.9843 | 0.9865 | 0.0022 | 0.9844 | 0.9864 | 0.002 |
| codrna | 0.9927 | 0.994 | 0.0013 | 0.9929 | 0.994 | 0.0011 |
| diabetes | 0.8086 | 0.8138 | 0.0052 | 0.7576 | 0.8123 | 0.0547 |
| fashion | 0.9901 | 0.9925 | 0.0024 | 0.9903 | 0.9925 | 0.0022 |
| ijcnn1 | 0.9941 | 0.9958 | 0.0017 | 0.9946 | 0.9962 | 0.0016 |
| mnist | 0.9992 | 0.9997 | 0.0005 | 0.9991 | 0.9997 | 0.0006 |
| mnist26 | 0.9995 | 0.9999 | 0.0004 | 0.9996 | 0.9999 | 0.0003 |
| sensorless | 1.0 | 1.0 | 0.0 | 1.0 | 1.0 | 0.0 |
| webspam | 0.9989 | 0.9992 | 0.0003 | 0.9989 | 0.9992 | 0.0003 |
| electricity | 0.9449 | 0.9714 | 0.0265 | 0.963 | 0.977 | 0.014 |
| drybean | 0.9947 | 0.9943 | -0.0004 | 0.9933 | 0.9951 | 0.0018 |
| adult | 0.9047 | 0.9158 | 0.0111 | 0.9215 | 0.934 | 0.0125 |
| banknote | 0.9958 | 0.9958 | 0.0 | 1.0 | 1.0 | 0.0 |
| voice | 0.987 | 0.9916 | 0.0046 | 0.9954 | 0.9986 | 0.0032 |
| waveform | 0.9731 | 0.975 | 0.0019 | 0.9567 | 0.9676 | 0.0109 |
| wind | - | - | - | - | - | - |
| speech | - | - | - | - | - | - |

Table 30: Cube RandomForest Experiments - ROC-AUC Degradation

| Dataset | Leaf-Tuple $L_2$ | | | Leaf-Tuple $L_\infty$ | | |
|---|---|---|---|---|---|---|
| | Splitted Dataset | Original Dataset | $\Delta$ROC-AUC | Splitted Dataset | Original Dataset | $\Delta$ROC-AUC |
| breast_cancer | 1.0 | 0.9975 | -0.0025 | 0.9969 | 0.9969 | 0.0 |
| covtype | - | - | - | - | - | - |
| codrna | 0.9927 | 0.994 | 0.0013 | 0.9928 | 0.994 | 0.0012 |
| diabetes | - | - | - | - | - | - |
| fashion | 0.99 | 0.9924 | 0.0024 | 0.9899 | 0.9925 | 0.0026 |
| ijcnn1 | 0.995 | 0.996 | 0.001 | 0.9949 | 0.9958 | 0.0009 |
| mnist | 0.9992 | 0.9997 | 0.0005 | 0.9992 | 0.9997 | 0.0005 |
| mnist26 | 0.9998 | 0.9999 | 0.0001 | 0.9997 | 0.9999 | 0.0002 |
| sensorless | 1.0 | 1.0 | 0.0 | 1.0 | 1.0 | 0.0 |
| webspam | 0.999 | 0.9992 | 0.0002 | 0.999 | 0.9992 | 0.0002 |
| electricity | 0.9546 | 0.9785 | 0.0239 | 0.948 | 0.9726 | 0.0246 |
| drybean | 0.9953 | 0.9959 | 0.0006 | 0.9942 | 0.9951 | 0.0009 |
| adult | 0.9211 | 0.9298 | 0.0087 | 0.9002 | 0.9208 | 0.0206 |
| banknote | 0.9966 | 1.0 | 0.0034 | 0.9974 | 1.0 | 0.0026 |
| voice | 0.9992 | 0.9986 | -0.0006 | 0.9941 | 0.9974 | 0.0033 |
| waveform | 0.9536 | 0.9677 | 0.0141 | 0.9359 | 0.9615 | 0.0256 |
| wind | 0.9249 | 0.9431 | 0.0182 | 0.9334 | 0.9356 | 0.0022 |
| speech | - | - | - | - | - | - |

Table 31: Leaf-Tuple RandomForest Experiments - ROC-AUC Degradation

# H APPENDIX H - METHOD FLOW - FURTHER EXPLANATIONS

Figure 28: The general flow of our method with a color legend at the bottom.

In Figure 28, you can see a visualization of the points in Subsection 4.1, which we added here below as well, for readers' convenience:

1. Split the dataset into four different parts for different purposes: $\mathcal{S}_{\mathcal{T}}$ to train the tree model, $\mathcal{S}_{\mathcal{E}}$ to train our basic representation model, $\mathcal{S}_{\mathcal{D}-train}$ to train our adversarial detector and $\mathcal{S}_{\mathcal{D}-test}$ to evaluate our adversarial detector.

2. Train a tree model.

3. Generate a triplets dataset that will be used to initialize the new representations. This is explained more fully in Subsection 4.2.

4. Train our basic embedding model $\mathcal{E}$. This is explained more fully in Subsection 4.3.

5. Split $\mathcal{S}_{\mathcal{D}-train}$ and $\mathcal{S}_{\mathcal{D}-test}$ into two parts.

6. Generate adversarial samples using an attack method $\mathcal{A}$.

7. Generate a new triplet dataset for each of the new sub-datasets. This is explained more fully in Subsection 4.4.

8. Optimize the representations of the new sub-datasets to our new embeddings using $\mathcal{E}$, more details in Subsection 4.4.

9. Concatenate the new representation of every set to the original ones.

10. Train our adversarial detector. This is explained more fully in Subsection 4.5.

11. Evaluate our adversarial detector. This is explained more fully in Section 5.

Our dataset is split into four main sub-datasets:

1. $\mathcal{S}_{\mathcal{T}}$ - a dataset that will be used to train the target tree model.

2. $\mathcal{S}_{\mathcal{E}}$ - a dataset that will be used to optimize a base set of embeddings that represent the general distribution of the original dataset. This base set of embeddings used to optimize new samples representations.

3. $\mathcal{S}_{\mathcal{D}-train}$ - a dataset used to train our adversarial detector, it will be split into two sub-sub-datasets to allow us to collect normal samples and adversarial samples.

4. $\mathcal{S}_{\mathcal{D}-test}$ - a dataset used to test our adversarial detector, it will be split into two sub-sub-datasets to allow us to collect normal samples and adversarial samples.

We chose this split method to reduce bias and overfitting when optimizing the representations and training our adversarial detector. Each of the above datasets is sampled randomly from the original dataset without replacements, meaning one sample will be only in one of the above sub-datasets. As a result, the amount of data we use for each one of the steps is reduced, and we added information about it in Appendix G to show how much it affects the classification performance of the original tasks.

