# OpenReview forum: "Adversarial Detector for Decision Tree Ensembles Using Representation Learning"
_ICLR.cc/2023/Conference — Submitted to ICLR 2023_

### Official Review · Reviewer_WNyf · 2022-10-22

**Confidence:** 3
**Correctness:** 3
**Technical Novelty And Significance:** 2
**Empirical Novelty And Significance:** 2
**Recommendation:** 5

**Clarity, Quality, Novelty And Reproducibility:**

The motivation and high level structure of the method is well presented. But the proposed method is kind of complicated and the rationale is not well explained and it seems that there are many hyperparameters, which may lead to some difficulties for reproduction.

**Strength And Weaknesses:**

Strength:
1. The authors conducted many experiments on different datasets to show the effectiveness of the proposed method.
Weakness:
1. In general, the proposed method is kind of over complicated. The authors split the original dataset into 4 parts and the tree model is only trained with S_T. Then the tree model itself's performance may be worse than before due to lack of data.
2. I think the rationale of model design in this paper is kind of unclear. More explanation and ablation studies are needed to show why each part is necessary. For example, in data representation, why we use the task in Algorithm 1 as representation? If the output is the whether s_i and s_j agree on the condition in n_k, are we simply predicting whether a specific feature's values of s_i and s_j are at the same side of n_k's threshold or not?
3. As the detector is a XGboost ensemble, it is also susceptible to adversarial attacks. If the attacker has access to this detector model, they may also be able to fool it and consequently fool the whole design.
4. I am concerned about the black box attacks used in the experiments and they may not be strong enough. I would suggest the authors to use the MILP method in Kantchelian et al 2016 as it is the strongest method for attacking tree ensembles (theoretically it is the optimal attack). An implementation of that method can be found in Chen et al 2019's Github Repo, which is compatible with XGBoost's data structure.

**Summary Of The Paper:**

This paper proposes a method to detect adversarial samples in tree ensembles without affecting either the model's structure or its original performance. Since the existing adversarial defense method may affect the model's natural performance, this paper's method enables the users to decided whether to apply defense or not. The proposed method extracts a new representation of a dataset based on the structure of the given tree ensemble to understand its behavior on normal examples and to detect adversarial examples. The authors conducted experiments to verify the effectiveness of the proposed method.

**Summary Of The Review:**

In general, I think the authors made a good amount of effort in design and experiments. But the proposed method is kind of too complicated and the rationale of this complex design is not well-explained. So I recommend a borderline reject.

---

> ### Author Response · Authors · 2022-11-15
> **Response to Reviewer WNyf**
>
> Thanks for reviewing our submission and providing us with constructive feedback. Please find our responses below:
> ***
> > In general, the proposed method is kind of over complicated. The authors split the original dataset into 4 parts and the tree model is only trained with S_T. Then the tree model itself's performance may be worse than before due to lack of data.
>
> The above is a valid point that we should have addressed. We added appendix H, which adds an explanation about the dataset splitting. Also, we compared the performance of our models to new models with the same hyperparameters, which we trained on all data besides the test set. We added appendix G to review and quantify how this affected the models' performance.
>
> ***
> > I think the rationale of model design in this paper is kind of unclear. More explanation and ablation studies are needed to show why each part is necessary. For example, in data representation, why we use the task in Algorithm 1 as representation? If the output is the whether s_i and s_j agree on the condition in n_k, are we simply predicting whether a specific feature's values of s_i and s_j are at the same side of n_k's threshold or not?
>
> Yes, that is precisely what we try to predict using the algorithm 1 dataset. We want to optimize a representation of each sample based on the tree structure and the route the sample passed through in the ensemble, considering the relevant features and thresholds.
>
> ***
> > As the detector is a XGboost ensemble, it is also susceptible to adversarial attacks. If the attacker has access to this detector model, they may also be able to fool it and consequently fool the whole design.
>
> Please see the comment to reviewer 9gBn.
> ***
>
> > I am concerned about the black box attacks used in the experiments and they may not be strong enough. I would suggest the authors to use the MILP method in Kantchelian et al 2016 as it is the strongest method for attacking tree ensembles (theoretically it is the optimal attack). An implementation of that method can be found in Chen et al 2019's Github Repo, which is compatible with XGBoost's data structure.
>
>
> Using also MILP is a valid point but, unfortunately, won’t be possible to achieve with the number of experiments and time constraints in the rebuttal; thus, we will keep it as a future work. However, MILP is a white box attack, and as mentioned in the comment for reviewer 9gBn, less likely to occur in a real-world scenario.
> ***
> > The motivation and high level structure of the method is well presented. But the proposed method is kind of complicated and the rationale is not well explained and it seems that there are many hyperparameters, which may lead to some difficulties for reproduction.
>
> Regarding the rationale issue, we explained our main contribution at the end of the introduction section.
> ***

---

### Official Review · Reviewer_yzr9 · 2022-10-22

**Confidence:** 3
**Correctness:** 3
**Technical Novelty And Significance:** 2
**Empirical Novelty And Significance:** 2
**Recommendation:** 3

**Clarity, Quality, Novelty And Reproducibility:**

Q1: Presentation quality is poor. The figure quality is very low, not up to the standard. The description of the algorithm, in particular, section 4.1 and 4.3 and Fig. 1 and 2, are very low. I cannot follow some parts of the algorithm description.

Q2: In terms of novelty, similar idea has been tested on DNN and has been broken.

Q3: I cannot follow the algorithmic description due to the quality of presentation. Thus, to me, it is not reproducible.

Q4: I recommend the authors asking a senior researcher to read and check their paper carefully.


**Details Of Ethics Concerns:**

N.A

**Strength And Weaknesses:**

Strength
1. The authors evaluate their algorithm on 18 databases with diverse sources and the databases have different properties, e.g., binary values and image etc,

2. The authors claim that their method is state-of-the-art.

weakness:
Q1)  The authors claim that “Most of the research is focused on adversarial attacks targeting neural network models because of the nature of their continuous learning space…” It is inaccurate statement. We study adversarial example against NN because 1) NN has been outperformed all other methods on many datasets, 2) Its performance is very high, in some cases, even outperforming human experts, 3) DNN has a lot of application and commercial values. However, decision trees are not comparable DNN in all these aspects.

Q2) The authors claim that “Tree-based models continue to be very popular (Nielsen 2016)”. 1) the reference is 6 years ago, and it is a master thesis. Please give better justification on this statement.

Q3) The authors mention that “Of the eleven datasets tested, seven shown a decrease in accuracy”. Decrease in accuracy is not necessary a problem. The question is how much.

Q4) Section 3,2 N1 is not defined although it is understandable.

Q5) Similar idea, extracting information from classifier to train an adversarial example detector has been proposed for DNN. They also have been broken. There is no theory to support that the proposed method is secure.

Q6) Since a lot of methods and tools have been developed for NN for classification, explanation and secure them, the necessity for using decision tree is weakened.



**Summary Of The Paper:**

The authors designed a method to detect adversarial examples against decision tree ensembles. The basic idea is to extract features from decision treed and use its representations on train a classifier to detect normal and adversarial example.

**Summary Of The Review:**

The presentation quality is not up to the standard, Many part is unclear.

There is not theory to support that the proposed method is secure.

---

> ### Author Response · Authors · 2022-11-15
> **Response to Reviewer yzr9**
>
> Thanks for reviewing our submission and providing us with constructive feedback. Please find our responses below:
> ***
> >Q1) The authors claim that “Most of the research is focused on adversarial attacks targeting neural network models because of the nature of their continuous learning space…” It is inaccurate statement. We study adversarial example against NN because 1) NN has been outperformed all other methods on many datasets, 2) Its performance is very high, in some cases, even outperforming human experts, 3) DNN has a lot of application and commercial values. However, decision trees are not comparable DNN in all these aspects.
>
> Thank you for that clarification; we appreciate this comment. We added an exception for the first claim that researchers choose to focus on adversarial attacks for NN *among other things* because.... .On a more general note, we don’t claim that decision trees outperform NN. As we also commented on the question below, boosting ensemble is still considered best performing on tabular data for several reasons [1] [2], which is why we believe it is worth researching a different solution for adversarial attacks on them.
>
> ***
>
> >Q2) The authors claim that “Tree-based models continue to be very popular (Nielsen 2016)”. 1) the reference is 6 years ago, and it is a master thesis. Please give better justification on this statement.
>
> We added a focus on tabular data and reference to two more recent works to justify this statement.
>
> ***
> >Q3) The authors mention that “Of the eleven datasets tested, seven shown a decrease in accuracy”. Decrease in accuracy is not necessary a problem. The question is how much.
>
> We did not review all tree robustness algorithms and did not quantify how much this “problem” is serious, we believe that its enough to highlight there is a potential problem and to define a new tool a model owner can use to hopefully protect their model without changing its performance. Even a slight decrease in accuracy might affect a critical solution and cost to some organizations using an ML model an extreme amount of money, thus, the possibility of allowing defending against adversarial attacks without damaging the model performance might be very important.
>
> ***
> >Q4) Section 3,2 N1 is not defined although it is understandable.
>
> Thank you for highlighting that, will be added in the rebuttal version.
>
> ***
> >Q5) Similar idea, extracting information from classifier to train an adversarial example detector has been proposed for DNN. They also have been broken. There is no theory to support that the proposed method is secure.
>
> Please refer to the comment for reviewer 9gBn
>
> ***
> >Q6) Since a lot of methods and tools have been developed for NN for classification, explanation and secure them, the necessity for using decision tree is weakened.
>
> There are different, more recent research focusing on attacks on decision trees based classifiers and ways to verify their robustness [3][4].  And as we mentioned above, we don’t claim decision trees are stronger than neural networks. Given the fact that they are still used, at least for tabular data (which different NN architectures have failed to outperform [1][2]), we believe this issue is worth researching.
>
> ***
>
> >Q1: Presentation quality is poor. The figure quality is very low, not up to the standard. The description of the algorithm, in particular, section 4.1 and 4.3 and Fig. 1 and 2, are very low. I cannot follow some parts of the algorithm description.
>
> > Q3: I cannot follow the algorithmic description due to the quality of presentation. Thus, to me, it is not reproducible.
>
> Thank you very much for highlighting this issue. We increased the resolution Fig 1 (now Fig 28), moved it to its own Appendix (Appendix H), and added an explanation about the dataset splitting. Fig 2 (now Fig 1) stayed in the same place, improving its resolution and making it clearer and readable.
>
> ***
>
> [1] Grinsztajn, Léo, Edouard Oyallon, and Gaël Varoquaux. "Why do tree-based models still outperform deep learning on tabular data?." arXiv preprint arXiv:2207.08815 (2022).
> [2] Shwartz-Ziv, Ravid, and Amitai Armon. "Tabular data: Deep learning is not all you need." Information Fusion 81 (2022): 84-90.
> [3] Zhang, Chong, Huan Zhang, and Cho-Jui Hsieh. "An efficient adversarial attack for tree ensembles." Advances in Neural Information Processing Systems 33 (2020): 16165-16176.
> [4] Vos, Daniël, and Sicco Verwer. "Efficient training of robust decision trees against adversarial examples." International Conference on Machine Learning. PMLR, 2021.

---

### Official Review · Reviewer_9gBn · 2022-10-24

**Confidence:** 4
**Correctness:** 2
**Technical Novelty And Significance:** 3
**Empirical Novelty And Significance:** 2
**Recommendation:** 3

**Clarity, Quality, Novelty And Reproducibility:**

The paper writing is not good and reads like some very rough draft. In fact, I feel that it does not read like a ``normal" ICLR paper I usually came across these days (no offense, just stating my feeling)

**Strength And Weaknesses:**

The topic seems novel - indeed that most of the current research focuses on vision research and thus deep nets.

On the other hand, from my understanding of this work, it does not seem to consider adaptive attacks -- namely adversarial attacks that takes into consideration of the proposed defense and then attack. For example, what happens that the attacker can try to generate attacks that bypass the proposed embedding network/detector?

**Summary Of The Paper:**

This paper studies adversarial robustness of decision trees, under the typical L_p norm attacks. The idea is to generate a representation data set, and then train a neural network to detect anomalies. Experiments seem to suggest that this approach works well.

**Summary Of The Review:**

The lack of considering adaptive attacks seems a major flaw, and I am not sure I am convinced by the experiments.

Also -- just that decision tree is used a lot does not necessarily justify that we should study the norm-based adversarial attacks for it (in vision, this motivation is very naturally justified), some real world examples why this is meaningful would help

---

> ### Author Response · Authors · 2022-11-15
> **Response to Reviewer 9gBn**
>
> Thanks for reviewing our submission and providing constructive feedback. Please find our responses below:
>
> ***
>
> >from my understanding of this work, it does not seem to consider adaptive attacks -- namely adversarial attacks that takes into consideration of the proposed defense and then attack. For example, what happens that the attacker can try to generate attacks that bypass the proposed embedding network/detector?
>
> The above is a valid point. There are mainly two settings relevant when considering adversarial attacks - a white and black box. When considering black box settings, the assumption is that the attacker has a queriable interface to the model, our method should not be a user-facing solution but rather an internal tool for the owner of the model, so there should not be an accessible queriable interface.
>
> In white box settings, which in general is less likely, our model might be a target of another layer of adversarial attacks. Nevertheless, our model merely allows the model owner to check if a sample is a possible attack without affecting the results of the target model, not preventing them entirely. Moreover, it reduces the system's attack surface because, with it, the attacker needs to generate an adversarial attack that will fool the model and, simultaneously, will be translated into an embedding vector that will fool our detector.
>
> On a more general note, we do not claim to obliterate adversarial attacks. We suggest a method to allow researchers and developers to increase the credibility of their model. This field is a constant chase between a cat and mouse; for example, until [1], the defense technique [2] was considered impenetrable and constant rivalry between the sides feeds the research progress. Even if our detector is not immune, it is a step towards detecting adversarial attacks without affecting the target model training procedure or its performance.
>
> ***
>
> >Also -- just that decision tree is used a lot does not necessarily justify that we should study the norm-based adversarial attacks for it (in vision, this motivation is very naturally justified), some real world examples why this is meaningful would help
>
> We agree it is an issue. That is why we did mention this point in our discussion section. Most of the notable decision trees targeted attacks, such as [3] [4] [5], are L_p attacks; some of them also on non-tabular datasets that we decided to use for our evaluation as well. There are works such as [6] [7] that we also mentioned in our paper that address this critical subject, but it still needs to be developed, and we decided not to increase the complexity of our work further. We will consider this an essential future work that needs to be reviewed and add a sentence about it in our conclusions in a revisioned version.
>
> ***
>
>
> [1] Carlini, Nicholas, and David Wagner. "Towards evaluating the robustness of neural networks." 2017 ieee symposium on security and privacy (sp). Ieee, 2017.
> [2] Papernot, Nicolas, et al. "Distillation as a defense to adversarial perturbations against deep neural networks." 2016 IEEE symposium on security and privacy (SP). IEEE, 2016.
> [3] Zhang, Chong, Huan Zhang, and Cho-Jui Hsieh. "An efficient adversarial attack for tree ensembles." Advances in Neural Information Processing Systems 33 (2020): 16165-16176.
> [4] Kantchelian, Alex, J. Doug Tygar, and Anthony Joseph. "Evasion and hardening of tree ensemble classifiers." International conference on machine learning. PMLR, 2016.
> [5] Zhang, Fuyong, et al. "Decision-based evasion attacks on tree ensemble classifiers." World Wide Web 23.5 (2020): 2957-2977.
> [6] Calzavara, Stefano, et al. "Treant: training evasion-aware decision trees." Data Mining and Knowledge Discovery 34.5 (2020): 1390-1420.
> [7] Vos, Daniël, and Sicco Verwer. "Efficient training of robust decision trees against adversarial examples." International Conference on Machine Learning. PMLR, 2021.

---

### Official Review · Reviewer_KhPV · 2022-10-25

**Confidence:** 5
**Correctness:** 3
**Technical Novelty And Significance:** 2
**Empirical Novelty And Significance:** 3
**Recommendation:** 5

**Clarity, Quality, Novelty And Reproducibility:**

Clarity:
This paper gives details of the algorithm step very lucidly.

Quality:
The quality of experiments is good. The technical depth however is a bit lacking.

Reproducibility:
The authors have given enough references and description to reproduce results.

Novelty:
It is unclear in the way the paper is presented, what non-obvious methods can be claimed as novelty. Stitching of existing known concepts or algorithms in a pipeline cannot be claimed as a main novelty - I expect a subsection listing of key contributions and hypothesis.

**Details Of Ethics Concerns:**

Not applicable. The docoloc plagiarism check is 6% which is excellent.

**Strength And Weaknesses:**

Strengths:
1. Table1 - the variety of datasets for experiments is appreciated
2. The appendix part gives more details on individual sub-section which is helpful
3. A different look away from crowd of NN works - traditional interpretable ML + tweaks put to good use
4. Good to see usage of UMAP instead of tSNE
5. Hard work has been put wrt the experiments

Weakness:
1. Logic behind feature threshold in n_k
2. Any logic for the 2-node setting in contrast to a m-node setting (seems like a mix of RANSAC and KNN)
3. Fig . 2 a - suddenly sigmoid came in pic - need figure explanation
4. Logic behind -  We extract four new sub-datasets
5. Embedding are not clear
6. Results need proper analysis - very little efoort put here
7. The abstract, conclusion should be to the point - this is the problem, what others have done and their gaps, this is what we have done with numbers to support.

**Summary Of The Paper:**

The paper describes a method to detect adversarial samples without affecting either the target model structure or its original performance. By using representation learning based on the structure of the tree ensembles, the claim is of better detection rates than SOTA. They claim the approach is better than using the original representation of the dataset to train an adversarial detector.

**Summary Of The Review:**

The paper in general needs some rework for ICLR  - please follow below.

Suggestions:
1.Rewrite the paper to bring in the main contributions to light and clearly explain the reseach gaps in SOTA where this paper is adding value.
2. Can use help of graph representations to explain the equation variables.
3. If the algorithm sub-modules can be mapped to graph theoritic standard problems, it will be good. Not sure about the time complexity of the task for large datasets and practical aspects.
4. Random node (sample) selection can be done smartly for better results.
5. Can some notes be written for explainability.
6. List down limitations of approach clearly

Miscellaneous:
1. The usage of XGBoost 2016 is a bit old - LightGBM (faster) and CatCBoost (more control) could have been tried.
2. Why numbering of lines is not done - difficult to review and refer the text for correction.
3. In this work, we present a technique to detect adversarial evasion attacks against tree-based classifiers and mainly boosting ensembles which heavily used. - reword - grammatically incorrect.
4. Unclear line - Our primary motivation for this work is to create a decision tree ensemble defense against adversarial
attacks that do not affect the model itself, allowing the model owner to decide if they want defense
applied to their existing model and fine-tune it.
5. For p ∈ N1 t , For p = ∞ --- format with commas for ease of reading
6. Fig 1, 2 font size is very small - need to zoom a lot - let it take the \linewidth (redraw if possible)
7. We define a supervised task like so: - grammar
8. Where are equation numbers? Hard to refer 0 use latex equation.
8. as our metrics; for multi-class cases - break the sentence please
9. Friedman test on the ROC-AUC, Nemneyi post-hoc test - explain

---

> ### Author Response · Authors · 2022-11-15
> **Response to Reviewer khPV**
>
> Thanks for reviewing our submission and providing us with constructive feedback. Please find our responses below:
>
> ***
>
> > Logic behind feature threshold in n_k
>
> In our method, there is no logic for choosing the threshold in n_k; we simply take the feature and threshold chosen inside the node from the target tree, which was chosen when the tree was constructed.
>
> ***
> > Fig . 2 a - suddenly sigmoid came in pic - need figure explanation
>
> Added a remark in 4.3.
>
> ***
> > Logic behind - We extract four new sub-datasets
>
> We added a new Appendix (Appendix H)  to elaborate more about our dataset splitting together with a better resolution figure of our method.
> ***
>
> >Embedding are not clear
>
> The main idea behind the embedding process is explained in 4.2.
> ***
> >Novelty: It is unclear in the way the paper is presented, what non-obvious methods can be claimed as novelty. Stitching of existing known concepts or algorithms in a pipeline cannot be claimed as a main novelty - I expect a subsection listing of key contributions and hypothesis.
>
> We elaborated on our main contributions at the end of the introduction section.
> ***
> > The usage of XGBoost 2016 is a bit old - LightGBM (faster) and CatCBoost (more control) could have been tried.
>
> The reason for using XGBoost was mainly its massive adoption in the industry and research and the evolved API and functionalities that the library allows us. Because the training of the tree model is not the heart of our work, but rather the adversarial samples, we chose to “choose different battle to confront with.”. However, we agree that a relevant future work will be to try this method with more boosting ensemble algorithms. Also relevant to mention is that we did not use the version of the library from 2016; we used the latest release.
>
> ***
> >In this work, we present a technique to detect adversarial evasion attacks against tree-based classifiers and mainly boosting ensembles which heavily used. - reword - grammatically incorrect.
>
> >Unclear line - Our primary motivation for this work is to create a decision tree ensemble defense against adversarial attacks that do not affect the model itself, allowing the model owner to decide if they want defense applied to their existing model and fine-tune it.
>
> >We define a supervised task like so: - grammar
>
> >Where are equation numbers? Hard to refer 0 use latex equation.
>
> >as our metrics; for multi-class cases - break the sentence please
>
> Very much appreciated. All are fixed in the revision.
> ***
> >Friedman test on the ROC-AUC, Nemneyi post-hoc test - explain
>
> Added an explanation in subsection 5.4
> ***

---

### Author Response · Authors · 2022-11-15
**Summary of Rebuttal changes**

We thank all the reviewers for investing their time, reading our submission, and providing constructive comments. We made the following changes:

1. Added Appendix G to review how much data reduction affected ML model performance due to our method.
2. Added Appendix H to make our flow sketch more readable and elaborate on our dataset split.
3. Rewording and grammar fixing (text in orange).
4. Added several missing comments (text in blue).
5. Added equations numbering.
6. Improved main figures' resolution and readability.

---

### Decision · Program_Chairs · 2023-01-20

**Decision:**

Reject

**Justification For Why Not Higher Score:**

N/A

**Justification For Why Not Lower Score:**

N/A

**Metareview: Summary, Strengths And Weaknesses:**

This research aims to create an adversarial detector for attacks on an ensemble of decision trees. And they demonstrate a method to detect adversarial samples without affecting either the target model structure or its original performance. They achieved better detection rates than using the original representation of the dataset to train an adversarial detector.

++ The positive part of this paper is the rich experiment (18 datasets).

-- All reviewers raised significant concerns about the current form of the paper, which is below the bar of ICLR.
- The writing of the paper does not meet ICLR standards. The abstract, method, and conclusion should be rewritten to clarify them.
- Many grammatical issues presented in the paper as pointed out by the reviewer.
- The motivation for combining an adversarial attack with a decision tree is not convincing.

The meta-reviewer recommends rejection.